



# Intense windstorms in the northeastern United States

**Frederick W. Letson**[1], **Rebecca J. Barthelmie**[2], **Kevin I. Hodges**[3], **and Sara C. Pryor**[1]

[1]Department of Earth and Atmospheric Sciences, Cornell University, Ithaca, New York, USA
[2]Sibley School of Mechanical and Aerospace Engineering, Cornell University, Ithaca, New York, USA
[3]Environmental System Science Centre, University of Reading, Reading, United Kingdom

**Correspondence:** Frederick Letson (fl368@cornell.edu) and Sara C. Pryor (sp2279@cornell.edu)

**Abstract.** Windstorms are a major natural hazard in many countries. The objective of this study is to identify and characterize intense windstorms during the last 4 decades in the US Northeast and determine both the sources of cyclones responsible for these events and the manner in which those cyclones differ from the cyclone climatology. The windstorm detection is based on the spatial extent of locally extreme wind speeds at 100 m height from the ERA5 reanalysis database. During the top 10 windstorms, wind speeds exceed their local 99.9th percentile over at least one-third of land-based ERA5 grid cells in this high-population-density region of the USA. Maximum sustained wind speeds at 100 m during these windstorms range from 26 to over 43 ms$^{-1}$, with wind speed return periods exceeding 6.5 to 106 years (considering the top 5 % of grid cells during each storm). Property damage associated with these storms, with inflation adjusted to January 2020, ranges from USD 24 million to over USD 29 billion. Two of these windstorms are linked to decaying tropical cyclones, three are Alberta clippers, and the remaining storms are Colorado lows. Two of the 10 re-intensified off the east coast, leading to development of nor'easters. These windstorms followed frequently observed cyclone tracks but exhibit maximum intensities as measured using 700 hPa relative vorticity and mean sea level pressure that is 5–10 times the mean values for cyclones that followed similar tracks over this 40-year period. The time evolution of wind speeds and concurrent precipitation for those windstorms that occurred after the year 2000 exhibit good agreement with in situ ground-based and remote sensing observations, plus storm damage reports, indicating that the ERA5 reanalysis data have a high degree of fidelity for large, damaging windstorms such as these. A larger pool of the top 50 largest windstorms exhibit evidence of only weak serial clustering, which is in contrast to the relatively strong serial clustering of windstorms in Europe.

## 1 Introduction

### 1.1 Hazardous wind phenomena

Hazardous wind phenomena span a range of scales from extra-tropical cyclones down to downburst and gust fronts associated with deep convection (Golden and Snow, 1991). Herein we focus on large-scale, long-duration "windstorms" associated with extra-tropical cyclones since they are likely to have the most profound societal impacts. These large-scale windstorms are a feature of the climate of North America and Europe and a major contributor to weather-related social vulnerability and insurance losses (Della-Marta et al., 2009; Feser et al., 2015; Hirsch et al., 2001; Changnon, 2009; Ulbrich et al., 2001; Haylock, 2011; Lukens et al., 2018; Marchigiani et al., 2013).

This analysis focuses on windstorms in the northeastern region of the United States as defined in the National Climate Assessment (USGCRP, 2018) (Table 1, Fig. 1a). The northeastern USA experiences a relatively high frequency of damaging storms, in particular during the cold season (Hirsch et al., 2001), and exhibits relatively high exposure due to both the large number of (i) highly populated, high-density urban areas (Fig. 1d, SEDAC, 2020; U.S. Census Bureau, 2019) and (ii) high-value (insured) assets. For example, New York state ranks 10th of the 50 US states in total direct economic losses related to natural hazards, with estimated losses of USD 12.54 billion in 2009 USD between 1960 and 2009 (Gall et al., 2011).

The northeastern states exhibit a very high prevalence of mid-latitude cyclone passages (Hodges et al., 2011; Ulbrich et al., 2009) and the associated extreme weather events (Bentley et al., 2019). They lie under a convergence zone of two prominent Northern Hemisphere cyclone tracks associated with cyclones that form or redevelop as a result of lee cyclogenesis east of the Rocky Mountains (Lareau and Horel, 2012). The first is associated with extra-tropical cyclones that have their genesis within/close to the US state of Colorado and typically track towards the northeast (Colorado lows, CLs) (Bierly and Harrington, 1995; Hobbs et al., 1996). The second is characterized by cyclones that have their genesis in/close to the Canadian province of Alberta and track eastwards across the Great Lakes (Alberta clippers, ACs). Alberta clippers generally move southeastward from the lee of the Canadian Rockies toward or just north of Lake Superior (Fig. 1a) before progressing eastward into southeastern Canada or the northeastern United States, with fewer than 10 % of the cases in the climatology tracking south of the Great Lakes (Thomas and Martin, 2007). The Great Lakes are known to have a profound effect on passing cyclones during ice-free and generally unstable conditions that prevail during September to November (Angel and Isard, 1997). Particularly during the early part of the cold season, cyclones that cross the Great Lakes are frequently subject to acceleration and intensification via enhanced vertical heat flux and low-level moisture convergence due to the lake–land roughness contrast (Xiao et al., 2018). Cyclones such as Alberta clippers that transit the Great Lakes during periods with substantial ice cover are subject to less alteration (Angel and Isard, 1997). The northeastern states are also impacted by decaying tropical cyclones (TCs) that track north from the Gulf of Mexico or along the Atlantic coastline (Baldini et al., 2016; Varlas et al., 2019; Halverson and Rabenhorst, 2013). Research on windstorm risk in Europe found that, although fewer than 1 % of cyclones that impact northern Europe are post tropical cyclones, they tend to be associated with higher 10 m wind speeds (Sainsbury et al., 2020). Tropical cyclones are also a major driver of extreme wind speeds along the US eastern seaboard (Barthelmie et al., 2021), and events such as Hurricane Sandy have been associated with large geophysical hazards in the US Northeast (Halverson and Rabenhorst, 2013; Lackmann, 2015). This region also experiences episodic nor'easters (NEs), extra-tropical cyclones that form or intensify off/along the US east coast and exhibit either a retrograde or northerly track, resulting in a strong northeasterly flow over the northeastern states (Hirsch et al., 2001; Zielinski, 2002).

There is evidence that intense winter wind speeds at the mid-latitudes have increased since 1950, due in part to increased frequency of intense extra-tropical cyclones (Ma and Chang, 2017; Vose et al., 2014). While long-term trends such as this from reanalysis products are subject to the effects of changing data assimilation (Bloomfield et al., 2018; Befort et al., 2016; Bengtsson et al., 2004), the 56-member

**Table 1.** Summary of the states that comprise the northeastern region as defined by the National Climate Assessment (USGCRP, 2018). State abbreviations and population from the 2010 US Census are also given (U.S. Census Bureau, 2019).

| Name | Abbreviation | 2010 population |
|------|------------|----------------|
| United States | US | 308 745 538 |
| Northeastern region | NE | 64 443 443 |
| Connecticut | CT | 3 574 097 |
| Delaware | DE | 897 934 |
| District of Columbia | DC | 601 723 |
| Maine | ME | 1 328 361 |
| Maryland | MD | 5 773 552 |
| Massachusetts | MA | 6 547 629 |
| New Hampshire | NH | 1 316 470 |
| New Jersey | NJ | 8 791 894 |
| New York | NY | 19 378 102 |
| Pennsylvania | PA | 12 702 379 |
| Rhode Island | RI | 1 052 567 |
| Vermont | VT | 625 741 |
| West Virginia | WV | 1 852 994 |

20th-century reanalysis exhibits a positive trend in the 98th-percentile wind speed over parts of the USA, including the northeastern states that are the focus of the current research (Brönnimann et al., 2012).

## 1.2 Socioeconomic consequences of windstorms

Economic losses associated with atmospheric hazards are substantial. Data from Munich Re indicate that annual "weather-related" losses at the global scale in 1997–2006 were USD 45.1 billion (inflation adjusted to 2006 USD) (Bouwer et al., 2007). In 2013, globally aggregated losses due to natural hazards were estimated at USD 125 billion (Kreibich et al., 2014). Data from the contiguous USA indicate 168 "billion-dollar disaster events" linked to atmospheric phenomena during 1980–2013 (Smith and Matthews, 2015). In the USA, three-quarters of total damages from natural hazards derive from hurricanes, flooding, and severe winter storms (including windstorms) (Gall et al., 2011). There is also evidence of a trend towards increasing economic impact from natural hazards within the USA even after adjusting for inflation. According to one report, "nationwide, annual losses rose from USD 4.7 billion in the 1960s to USD 6.7 billion in the 1970s, USD 7.6 billion in the 1980s, USD 14.8 billion in the 1990s, and USD 23.6 billion in the 2000s" due to a combination of more frequent disasters, disasters of larger scale, and changes in societal resilience (Gall et al., 2011).

Windstorms present a hazard to the built environment; transportation, especially to aviation (Young and Kristensen, 1992); and multi-energy systems, including the electric grid (Bao et al., 2020; Wanik et al., 2015). In 2016 the annual cost of grid disruptions within the USA was estimated to

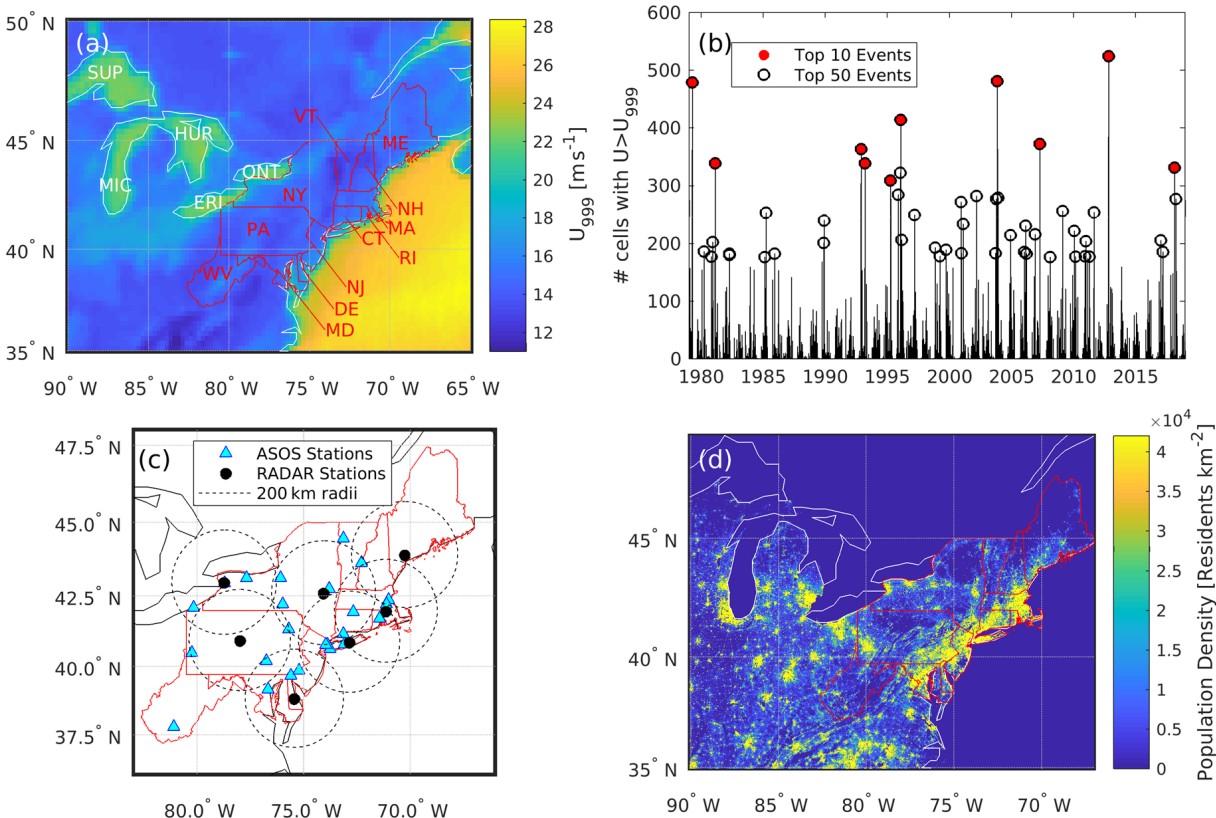

**Figure 1. (a)** 99.9th-percentile wind speed ($U_{999}$) from ERA5 for each grid cell in the northeastern USA derived using hourly wind speeds at 100 m a.g.l. during 1979–2018. Borders of the 12 northeastern states are shown in red. The Great Lakes are each labeled in white, with the first three letters of their names: Superior (SUP), Michigan (MIC), Huron (HUR), Erie (ERI), and Ontario (ONT). **(b)** Time series of the number of ERA5 grid cells over the northeastern states that exceed their local $U_{999}$ value (out of 924 cells). The 50 largest-magnitude events are circled in black, and the top 10 events are marked in red. **(c)** Locations of the 24 ASOS stations and 7 radar stations used for validation of ERA5 wind speed and precipitation values. The dotted circles show the area with 200 km radius from each radar station. **(d)** Population density of the Northeast at a spatial resolution of 30 arcsec ($\sim$ 1 km; data from the 2010 US Census available from the Socioeconomic Data and Applications Center (SEDAC, 2020)).

range from approximately USD 28 billion to USD 209 billion (Mills and Jones, 2016). Composite events characterized by the co-occurrence of ice accumulation and wind are particularly hazardous to the built environment, aviation, and energy infrastructure (Sinh et al., 2016; Jeong et al., 2019). For example, in the 1998 northeastern ice storm ice deposition combined with high winds led to the toppling of 1000 transmission towers, loss of power to 5 million people, and 840 000 insurance claims valued at USD 1.2 billion (Mills and Jones, 2016). This work seeks to advance understanding of the character and causes of extreme windstorms in the Northeast.

## 1.3 Objectives of this research

This research is inspired by and is conceptually analogous to development of the XWS (eXtreme WindStorms) catalogue of storm tracks and wind gust footprints for 50 of the most extreme European winter windstorms (Roberts et al., 2014).

Specific goals of the research reported herein are to do the following:

1. Present a new method for identifying and physically characterizing severe windstorms. This method is applied to 40 years of hourly output from the ERA5 reanalysis to extract the 10 most intense windstorms over the US northeastern states and describe them in terms of their location, spatial extent, duration, and severity. We further evaluate the degree to which these windstorms are composite extreme events, wherein high wind speeds co-occur with extreme or hazardous precipitation.

2. Verify aspects of the windstorms as characterized based on ERA5 reanalysis output using wind speed observations from sonic anemometers and precipitation characteristics from radar and in situ rain gauges, plus storm damage reports.

3. Contextualize these windstorms in the long-term cyclone climatology. Specifically, we track each windstorm over time and space using two indices of intensity derived from mean surface pressure and relative vorticity and compare their location and intensity to those of all cold-season cyclones affecting the northeastern USA from 1979 to 2018.

4. Evaluate these windstorms in terms of the return periods (RPs) of extreme wind speeds derived using the Gumbel distribution applied using annual maximum wind speeds for 1979–2018.

This research is a part of the HyperFACETS project, which uses a storyline-based analysis framework. Storylines are "physically self-consistent unfolding of past events, or of plausible future events or pathways" (Shepherd et al., 2018). They provide a method of framing a research inquiry in terms of three elements: a geographic region, a historically important or notable event, and a set of process drivers for that event.

## 2 Data and methods

### 2.1 ERA5 reanalysis

Attempts to identify and characterize windstorms from a geophysical perspective and contextualize them in a climatological setting have historically been hampered by limited data availability and/or quality from geospatially inhomogeneous observing networks. Further, time series from in situ wind measurement networks exhibit substantial inhomogeneities due to factors such as station relocations, instrumentation changes, changes in conditions around individual measurement stations, and changes in measurement frequencies and/or integration periods (Pryor et al., 2009; Wan et al., 2010). Thus, herein we employ once-hourly wind speeds from the ERA5 reanalysis. The wind speeds are for a height of 100 m a.g.l. at the model time step of $\sim 20$ min and a spatial resolution of $0.25° \times 0.25°$. This study focuses on windstorms within a study domain that extends from 35 to 50° N and 65 to 90° W (Fig. 1a). The events are defined using data from the 924 ERA5 land-dominated grid cells over the 12 northeastern states (two-letter abbreviations given in Table 1).

The ERA5 reanalysis is derived using an unprecedented suite of assimilated in situ and remote sensing observations (Hersbach et al., 2020). It exhibits relatively high fidelity for wind speeds (Kalverla et al., 2019, 2020; Olauson, 2018; Pryor et al., 2020; Jourdier, 2020; Ramon et al., 2019). However, it is important to acknowledge that wind parameters from any model do not fully reflect all scales of flow variability (Skamarock, 2004) and underestimate extreme wind speeds (Larsén et al., 2012), particularly in areas with high orographic complexity and/or varying surface roughness length. Here we use wind speeds at 100 m height because the events we seek to characterize are on regional rather than local scale and are necessarily driven by winds aloft. Flow at this height is less likely to be impacted by sub-grid-scale heterogeneity in surface roughness length and uncertainties induced by unresolved sub-grid scale variability. Near-surface wind speeds are strongly coupled to wind speeds at 100 m (i.e., within the PBL), but wind speeds at 100 m are less strongly impacted by inaccuracies and/or uncertainty in surface roughness length ($z_0$) (Minola et al., 2020; Nelli et al., 2020). Applying an uncertainty of a factor of 2 to $z_0$ can lead to mean differences of up to $0.75\,\mathrm{ms^{-1}}$ for near-surface (40 to 150 m a.g.l.) wind speeds (Dörenkämper et al., 2020). Estimates of wind gusts at a nominal height of 10 m are generated as a post-processing product from the ERA5 reanalysis product using the sustained wind speed at 10 m along with a term representing shear stress and a convective term (Minola et al., 2020). The association between these wind gust estimates and sustained wind speeds at 100 m are also presented and provide a link to previous research on European windstorms that focuses on wind gusts.

Cyclone tracking and intensity estimates presented herein employ 3-hourly mean sea level pressure (MSLP) and relative vorticity at 700 hPa (RV) fields from ERA5. Previous research has indicated relatively good consistency between cyclone climatologies derived using ERA5 and other recent reanalyses (Gramcianinov et al., 2020; Sainsbury et al., 2020). RV values at 700 hPa are used rather than 850 hPa as in the XWS European analysis due to the presence of high-elevation areas in US cyclone source regions. Further, the 3-hourly fields from ERA5 used herein are direct products of the reanalysis, whereas the 3-hourly values used in XWS were based on 6-hourly ERA Interim reanalyses combined with ERA Interim forecast values for the intervening time steps (Roberts et al., 2014).

Compound events, windstorms which exhibit a co-occurrence of extreme precipitation and/or freezing rain with high winds, are associated with amplified risk (Zscheischler et al., 2018; Sadegh et al., 2018). Precipitation intensity and hydrometeor class from ERA5 are used to identify to what degree each of the 10 windstorms identified here are compound events. The hydrometeor classes reported by ERA5 are rain, mixed rain and snow, thunderstorms, wet snow, dry snow, freezing rain, and ice pellets and are differentiated based largely on the temperature structure in the reanalysis model (https://confluence.ecmwf.int/display/FUG/9.7+Precipitation+Types, last access: 2 June 2021). Prior analyses of ERA5 precipitation values have indicated skill relative to in situ observations and gridded datasets over the USA (Tarek et al., 2020; Sun and Liang, 2020).

### 2.2 Observational data

Wind speeds and precipitation characteristics during the windstorms are identified using ERA5 and are validated us-

ing in situ measurements from 24 National Weather Service (NWS) Automated Surface Observation System (ASOS) stations and seven NWS radars (Fig. 1c). Since major upgrades to the NWS systems were conducted in 2000, this evaluation is focused on windstorms that occurred after that year. Five-minute measurements of in situ wind speeds at 10 m a.g.l. used in this evaluation derive from ice-free two-dimensional sonic anemometers (Schmitt IV, 2009), while the in situ observations of precipitation intensity reported from the ASOS network derive from heated tipping-bucket rain gauges (Tokay et al., 2010). In the absence of widespread in situ wind speed observations from tall towers (which would be more comparable to the 100 m wind speeds from ERA5), these 10 m wind speed observations represent the best available validation dataset for the occurrence of high winds throughout the Northeast states. NWS protocols document accumulated precipitation since the last hour, sampled every minute and reported every 5 min (Nadolski, 1998). For the current comparison to ERA5, these are averaged to generate hourly rainfall rates.

Precipitation rates from seven NWS dual-polarization radars (Kitzmiller et al., 2013) are used to provide an areally averaged comparison of ERA5 (Fig. 1c). NWS radar precipitation products are the result of extensive development efforts (Cunha et al., 2015; Villarini and Krajewski, 2010; Straka et al., 2000) and have been employed in a wide array of applications (Letson et al., 2020; Seo et al., 2015; Krajewski and Smith, 2002). Precipitation intensity rates derived from radar reflectivity are reported in 41 400 cells using 1° azimuth angle and a range resolution of 2 km. In the current work, precipitation rates over the land areas of northeastern states from radar and ASOS and ERA5 that are within 200 km of the seven radars are averaged in time to match the hourly resolution of ERA5 precipitation and interpolated in space to the 0.25° × 0.25° ERA5 grid (Fig. 1c).

## 2.3 NOAA Storm Events Database

The US National Oceanic and Atmospheric Administration (NOAA) provides detailed information on "the occurrence of storms and other significant weather phenomena having sufficient intensity to cause loss of life, injuries, significant property damage, and/or disruption to commerce" at the county level in the NOAA Storm Events Database (https://www.ncdc.noaa.gov/stormevents/, last access: 10 February 2021). These records are subject to some inhomogeneities associated with digitization of transcripts prior to 1993 and standardized into 48 event types in 2013 (https://www.ncdc.noaa.gov/stormevents/details.jsp?type=collection, last access: 10 February 2021). They are compiled from a range of county, state, and federal agencies in addition to the NWS. Like all hazard loss datasets they are subject to reporting inaccuracies and inconsistencies (Gall et al., 2009), but they represent a long and relatively consistent record and are

widely used (Young et al., 2017; Konisky et al., 2016). Damage and mortality estimates from this dataset provide an estimate of the impact of each windstorm, with the caveat that population density and hence the potential for loss of life and damage vary markedly between US counties that also vary greatly in area (Fig. 1d).

## 2.4 Method used to characterize windstorms

A range of different techniques have been developed and applied to identify and characterize atmospheric hazards including extreme windstorms. Some rely on an assessment of event severity such as insured losses or human mortality/morbidity. Others prescribe a level of rarity (i.e., they are probabilistic), while others prescribe a level of intensity (i.e., the occurrence of extreme values of some physical phenomena) (Stephenson, 2008). Here we employ a methodology based on the intensity and spatial extent of extreme wind speeds. This approach is conceptually similar to storm severity indices derived from European work based on the maximum 925 hPa wind speed within a 3° radius of the vorticity maximum and the area over which wind speeds at that height exceed $25 \, \mathrm{ms^{-1}}$ (Roberts et al., 2014; Della-Marta et al., 2009), while the current work considers over-threshold winds within a fixed domain 15 × 25° in extent. It also draws from earlier work that used an index defined as the product of the cube of the maximum observed wind speed over land, the areas impacted by damaging winds ($>25.7 \, \mathrm{ms^{-1}}$), and the duration of damaging winds (Lamb, 1991).

This analysis employs hourly wind speeds at 100 m a.g.l. for 1979–2018 in all 924 land-dominated grid cells over the northeastern states. The methodology applied to identify and characterize the 10 largest windstorms does not employ an absolute threshold of wind speed, but rather exceedance of locally determined thresholds defined by the 99.9th-percentile wind speed value ($U_{999}$). A local $U_{999}$ threshold is used, rather than an absolute wind speed threshold in meters per second, in part because storms affecting urban areas, which may not be prone to high wind speeds, may still result in damage to infrastructure. While lower percentile thresholds have been used in previous work (Walz et al., 2017; Klawa and Ulbrich, 2003), use of the 99.9th-percentile wind speed value is appropriate for identifying the truly extraordinary conditions we seek to characterize and is robust when applied to very long datasets with very large sample sizes. Use of locally determined thresholds also enables direct comparison of the spatial scale and intensity of windstorms derived using the ERA5 data at 100 m a.g.l. and near-surface wind speed observations from 10 m a.g.l. Exceedance of the local 99.9th-percentile wind speed value ($U_{999}$) is considered in both cases based on the ∼ 20-year record from each ASOS station and the 40 years of ERA5 data, and comparisons are made at an hourly resolution by averaging all ASOS wind speeds within a given hour.

**Table 2.** TS1 Summary of the top 10 windstorms listed in rank order of spatial extent. The time of max coverage ($t_p$) shows the time (in UTC) and date (listed as year/month/day) with the greatest geographic extent of high wind speeds. No. cells indicates the count of ERA5 grid cells (out of 924) with $U>U_{999}$ at $t_p$. The maximum precipitation accumulated in any northeastern state land grid cell is given for the 24 h surrounding the storm peak. Maximum sustained wind speeds at 100 m ($U$) and wind gusts ($G_{10}$) at 10 m are given for the 924 northeastern state land grid cells during each storm, for both $t_p$ and the entire wind storm period (97 h). Property damage for the northeastern states is based on NOAA storm damage reports and is accumulated over the duration of the period for which the associated cyclone (defined using RV) is evident. Inflation adjusted property damage is derived using inflation estimates from the US Bureau of Statistics (https://www.bls.gov/data/inflation_calculator.htm, last access: 2 June 2021). For comparative purposes, results from an analysis using a 98th-percentile wind speed threshold are shown in the final two columns. $U>U_{98}$ storm rank denotes the rank of windstorms defined using that local threshold, and no. cells $U>U_{98}$ indicates the number of NE grid cells that exceed their local 98th-percentile value.

| Time of max coverage ($t_p$) | No. cells $U>U_{999}$ | Max $U$ at $t_p$ [ms$^{-1}$] | Max $U$ during storm period [ms$^{-1}$] | Max $G_{10}$ at $t_p$ [ms$^{-1}$] | Max $G_{10}$ during storm period [ms$^{-1}$] | Max 24 h precip [mm] | Property damage [M USD] | Property damage [M USD] inflation adjusted to January 2020 | $U>U_{98}$ storm rank | No. cells $U>U_{98}$ |
|---|---|---|---|---|---|---|---|---|---|---|
| 2012/10/18 09:00 | 524 | 34.27 | 41.8 | 34.44 | 42.43 | 146.03 | 25 304 | 29 100 | 3 | 864 |
| 2003/11/11 00:00 | 481 | 26.04 | 29.95 | 36.58 | 37.18 | 39.02 | 1119 | 1600 | 29 | 717 |
| 1979/4/4 00:00 | 479 | 28.53 | 31.88 | 31.98 | 33.99 | 34.19 | 586 | 2233 | 34 | 697 |
| 1996/1/26 00:00 | 414 | 25.76 | 30.81 | 29.69 | 37.02 | 60.64 | 1298 | 2181 | 2 | 876 |
| 2007/4/11 21:00 | 372 | 29.56 | 32.44 | 31.04 | 34.07 | 79.06 | 392 | 502 | 24 | 729 |
| 1992/11/12 21:00 | 363 | 25.53 | 28.34 | 30.4 | 31.94 | 54.01 | 42 | 79 | 5 | 838 |
| 1981/2/11 00:00 | 339 | 24.81 | 29.08 | 27.66 | 36.61 | 93.02 | 8 | 24 | 20 | 746 |
| 1993/3/12 06:00 | 339 | 40.95 | 43.15 | 34.38 | 38.49 | 84.33 | 34 | 62 | 12 | 806 |
| 2018/3/1 03:00 | 331 | 31.66 | 33.1 | 33.77 | 35.39 | 84.71 | 164 | 172 | 48 | 641 |
| 1995/4/4 15:00 | 309 | 24.21 | 26.29 | 32.96 | 32.96 | 19.19 | 225 | 389 | 114 | 511 |

As shown in Fig. 1a, there is marked spatial variability in the 99.9th-percentile wind speed (i.e., the wind speed exceeded for slightly over 3500 h during the 40-year period). $U_{999}$ ranges from over 28 ms$^{-1}$ over the Atlantic Ocean down to 12 ms$^{-1}$ over some land grid cells due to the higher surface roughness and topographic drag. Windstorms are identified as periods when the largest number of ERA5 grid cells exceed their local (ERA5 grid-cell-specific) 99.9th-percentile wind speed value ($U>U_{999}$). A further restriction is applied in that no event may be within 14 days of any other, to avoid double counting of any individual storm (Fig. 1b, Table 2).

The peak hour of $U>U_{999}$ coverage within the Northeast states for each of the 10 most intense storms is referred to herein as the peak windstorm time ($t_p$), and the 97 h including and surrounding ($\pm$48 h) $t_p$ is referred to as the storm period. For each hour of each storm period a high-wind centroid is identified using the mean latitude and longitude of all grid cells where $U>U_{999}$.

Precipitation associated with each of the 10 most intense windstorms is also evaluated using ERA5 precipitation totals and types. The analysis of precipitation focuses on a 24 h period centered on the peak windstorm time ($t_p$). Precipitation statistics including 24 h total precipitation, hourly precipitation rates, and the frequency of each precipitation type are characterized for all land grid cells in northeastern states that exceed their local $U_{999}$ value at any point in this 24 h period.

Research from Europe indicates evidence of serial clustering of windstorms (Walz et al., 2018). Although our focus is primarily on the 10 most intense and extensive windstorms, a larger sample of 50 events is extracted using the methodology described above but relaxing the temporal separation from 14 to 2 d, to examine the degree to which spatially extensive windstorms over the Northeast as manifest in ERA5 are serially clustered (Fig. 1b). This analysis employs a Poisson distribution fit to the annual occurrence rate for these 50 events and the dispersion index ($D$) of (Mailier et al., 2006)

$$D = \frac{\sigma^2}{\mu} - 1, \tag{1}$$

where $\sigma^2$ and $\mu$ are the variance and mean of the distribution of the annual rates of occurrence. For a Poisson distributed random variable $\sigma^2$ and $\mu$ are equal (Wilks, 2011a). $D>0$ indicates the presence of temporal clustering. The significance of $D$ is evaluated using a bootstrapping analysis in which 10 000 samples are drawn with replacement and the dispersion index is calculated for each, similar to a method used in Pinto et al. (2016).

## 2.5 Development of a cyclone climatology

A cyclone detection and tracking algorithm (Hodges et al., 2011) is applied to 3-hourly ERA5 MSLP and 700 hPa RV global fields that have been subjected to T42 spectral filtering

for RV (corresponding to a 310 km resolution at the Equator) and T63 filtering for MSLP (210 km at the Equator) with the large-scale background removed for total wavenumbers $\leq 5$. These spectral filters are designed to restrict detection to tropical and mid-latitude cyclones (Hoskins and Hodges, 2002). The location and intensity of the cyclones are identified using the local maxima in RV and the minima (i.e., negative deviations) in MSLP relative to the filtered fields. RV cyclone intensities are shown in units of $10^{-5}\,\mathrm{s}^{-1}$, and MSLP intensity estimates are given in hectopascals scaled by $-1$. These anomalies are relative to removal of the large-scale background for $n \leq 5$, where $n$ is the total wavenumber in the spherical harmonic representation of the field. The cyclones are tracked by first initializing a set of tracks based on a nearest-neighbor method which are then refined by minimizing a cost function for track smoothness as in the XWS European analysis (Roberts et al., 2014). Cyclones only contribute to the climatology if they persist for $\geq 8$ time steps (24 h). The cyclone detection algorithm is applied separately to MSLP and RV, with the results being used to provide a qualitative assessment of the uncertainty in the cyclone tracks.

Tracks associated with each windstorm are identified from the geographic centroid of ERA5 grid cells where $U > U_{999}$ and secondly from the local maximum of MSLP (scaled by $-1$) and RV anomalies that tracked into the Northeast study domain during the storm period. The date and location at which the cyclone associated with each windstorm is first identified by the tracking algorithm are used to identify the source area of each windstorm, and the location and time at which the detection algorithm ceases to identify a cyclone are used to describe the end of the cyclone track. Subjective evaluation of the cyclone tracks associated with each windstorm is used to identify the type of cyclone associated with each windstorm. A cyclone is identified as an AC if the cyclone track originates over the North American continent north of $40°\,\mathrm{N}$, as a CL if the track originates over the North American continent south of $40°\,\mathrm{N}$, and as a decaying TC if the track originates south of $30°\,\mathrm{N}$ over a water grid cell. The term nor'easters is applied if the cyclone retrogrades towards the coastline after moving offshore and/or is associated with strong northeasterly flow over the northeastern states.

Consistent with past research (Hirsch et al., 2001), all of the top 10 windstorms identified using the largest spatial extent of locally extreme wind speeds in the ERA5 data occur during cold-season months (October to April). Thus, the cyclone track density used to contextualize the windstorms is restricted to only those months. This analysis further focuses solely on cyclones that track into the northeastern domain. These restrictions allow direct evaluation of the degree to which the windstorms are typical of the prevailing cyclone climatology.

## 2.6 Calculation of long-term period wind speeds

Peak wind speeds ($U_{\mathrm{peak}}$) during each of the windstorms are expressed in terms of their RP (in years) to provide a metric of the degree to which these events are exceptional. These statistics are computed for each ERA5 grid cell by fitting a double exponential (Gumbel) distribution to annual maximum wind speeds ($U_{\mathrm{max}}$) (Mann et al., 1998):

$$P(U_{\mathrm{max}}; \alpha, \beta) = e^{-e^{-(U_{\mathrm{max}} - \alpha)/\beta}}, \tag{2}$$

where the distribution parameters $\alpha$ and $\beta$ are derived using maximum-likelihood estimation. The $U_{\mathrm{peak}}$ estimates for each ERA5 grid cell are then evaluated in terms of their RP (in years) using (Wilks, 2011a; Pryor et al., 2012)

$$\mathrm{RP} = \frac{1}{1 - P(U_{\mathrm{peak}})}. \tag{3}$$

This method is similar to that used for grid-point-based wind speed return period calculations in previous work (Della-Marta et al., 2009), which resulted in return periods of 0.1 to 500 years when considering 200 prominent windstorms in Europe.

Uncertainty intervals in the return period wind speeds are assigned using the 95 % confidence intervals on the $\alpha$ and $\beta$ parameters as derived using maximum-likelihood estimation.

## 2.7 Loss index

Previous research has advocated use of a loss index (LI) to identify societally relevant wind storms (Klawa and Ulbrich, 2003):

$$\mathrm{LI} = \sum_{\mathrm{NE\,grid\,cells}} \mathrm{pop(cell)} \left( \frac{U_{\mathrm{max}}(\mathrm{cell})}{U_{98}(\mathrm{cell})} - 1 \right)^3, \tag{4}$$

where pop(cell) is the population of a reanalysis grid cell; $U_{\mathrm{max}}$ is the 24 h maximum wind speed in that grid cell; and $U_{98}$ is the local, long-term 98th-percentile wind speed. Here we evaluate the degree of correspondence between this LI applied here to wind speeds at 100 m and NOAA storm damage reports using linear fitting with zero intercept. Variance explanation ($R^2$) values for fits with forced zero intercept is computed using

$$R^2 = \frac{\sum \widehat{Y_i^2}}{Y_i^2}, \tag{5}$$

where $\widehat{Y_i^2}$ is the estimated value of damage ($Y$) for each storm ($i$) and $Y$ is the observed value for that event (Eisenhauer, 2003) from NOAA storm damage reports.

## 3 Results

### 3.1 Windstorm identification and characterization

The top 10 windstorms during 1979–2018 over the northeastern states identified using the method described above

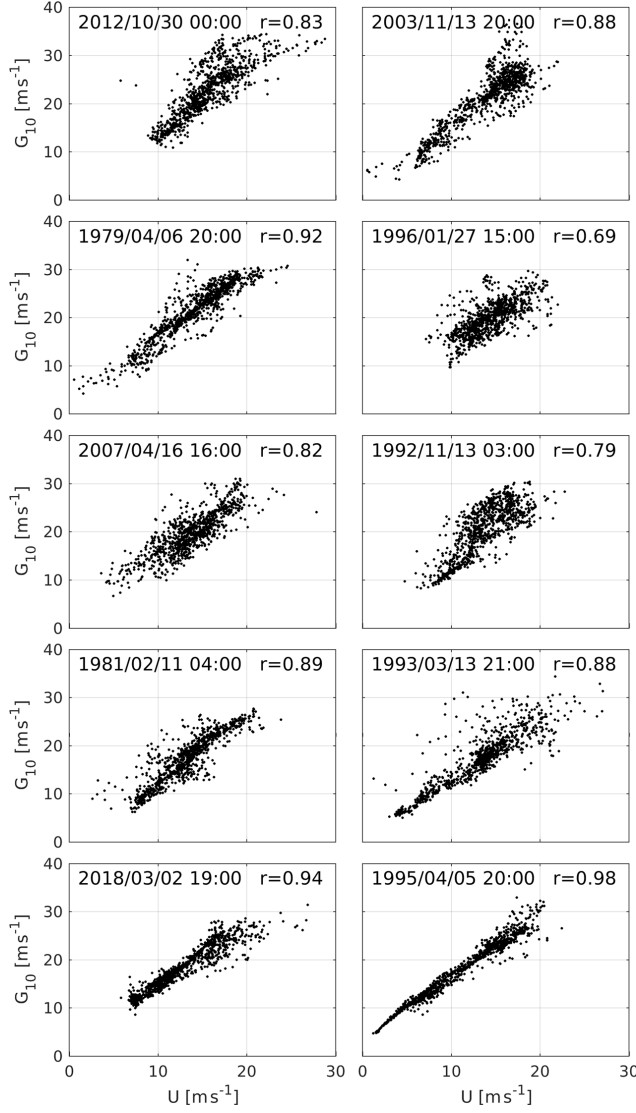

**Figure 2.** Maximum gust at 10 m a.g.l ($G_{10}$) vs. hourly 100 m wind speed ($U$) in all Northeast grid cells during the peak hour ($t_p$) of each of the top 10 storms. The spatial correlation coefficient ($r$) is also given for each storm.

The maximum wind speed at 100 m a.g.l. in any ERA5 grid cell at the peak hour ranges from 25 to 41 ms$^{-1}$, while the maximum during the storm period ranges from 26 to 44 ms$^{-1}$ (Table 2). These maximum wind speeds do not scale with the storm intensity as measured by the number of grid cells that exceed their local 99.9th-percentile wind speeds (Table 2). For example, the windstorm during March 1993 is associated with the highest absolute wind speeds but is manifest in a relatively small number of ERA5 grid cells. Maximum wind gusts at 10 m a.g.l. ($G_{10}$) exceed the sustained wind speeds at 100 m a.g.l. at both the peak hour and over the entire windstorm. Maximum $G_{10}$ from ERA5 for all windstorms is well above the US National Weather Service "damaging winds" threshold of 25.7 ms$^{-1}$ (Trapp et al., 2006) (Table 2). The spatial correlation coefficient between 100 m sustained wind speeds and $G_{10}$ at $t_p$ is $>0.68$ for all storms and $>0.8$ for 8 out of the 10 storms, indicating that the 100 m sustained wind speeds analyzed herein are strongly related to near-ground wind gusts in the ERA5 reanalysis (Fig. 2).

All 10 windstorms are associated with substantial damage reports within the Northeast states (Table 2, Fig. 3), and 9 of the 10 storms were responsible for deaths in the Northeast states (Fig. 3). There is not direct correspondence between the ranking of the windstorms in terms of the number of ERA5 grid cells with $U > U_{999}$ and the amount of damage and human mortality as reported in the NOAA storm data, but the four highest-magnitude windstorms (2012, 2003, 1979, and 1996; i.e., those ranked 1–4) all have property damage totals above any of the other six windstorms (Table 2). Further, although NOAA storm data indicate only modest total economic costs associated with property damage during the 1992 windstorm, there are reports of widespread damage in counties across much of the Northeast (Fig. 3). The lack of complete correspondence between the centroid of windstorms, as identified using the methodology presented here, and property damage in the NOAA dataset is likely due to the following: (i) occurrence of localized extreme (damaging) winds that are manifest at scales below those represented in the ERA5 reanalysis (e.g., downbursts from embedded thunderstorms, sting jets, and other mechanisms; Li et al., 2020; Clark and Gray, 2018) (a grid resolution of 20 km or higher may be required to fully capture damaging winds; Hewson and Neu, 2015); (ii) spatial variability in insured assets (Nyce et al., 2015; Brown et al., 2015); (iii) possible inconsistences in storm-reporting practices across counties (see NOAA storm data publications for details: https://www.ncdc.noaa.gov/IPS/sd/sd.html, last access: 10 February 2021); and (iv) compound events involving heavy precipitation, icing, or storm surge (e.g., Hurricane Sandy; Wang et al., 2014), along with intense winds, which may be associated with increased damage. Nevertheless, although many factors dictate economic losses from windstorms, the Pearson correlation coefficient ($r$) between the number of grid cells with $U > U_{999}$ at $t_p$ and inflation-adjusted property damage exceeds 0.66, and $r$ between the maximum wind speed

are summarized in Table 2. During the peak hour ($t_p$) of each of these windstorms, 309 to 524 (33 % to 56 %) of the 924 ERA5 land-dominated grid cells exhibit $U > U_{999}$ (Table 2). For context, 10 % of ERA5 grid cells co-exhibit $U > U_{999}$ in $<1$ % of hours. The windstorms are not concentrated in any sub-period of the 40 years under consideration (1979–2018), and no individual year contained 2 of the top 10 windstorms (Fig. 1b). Hence, in the following the windstorms are referred to by their (unique) year of occurrence, and in all figures and tables results are displayed in decreasing order of windstorm magnitude as defined using the spatial extent of $U > U_{999}$ at $t_p$ (Table 2).

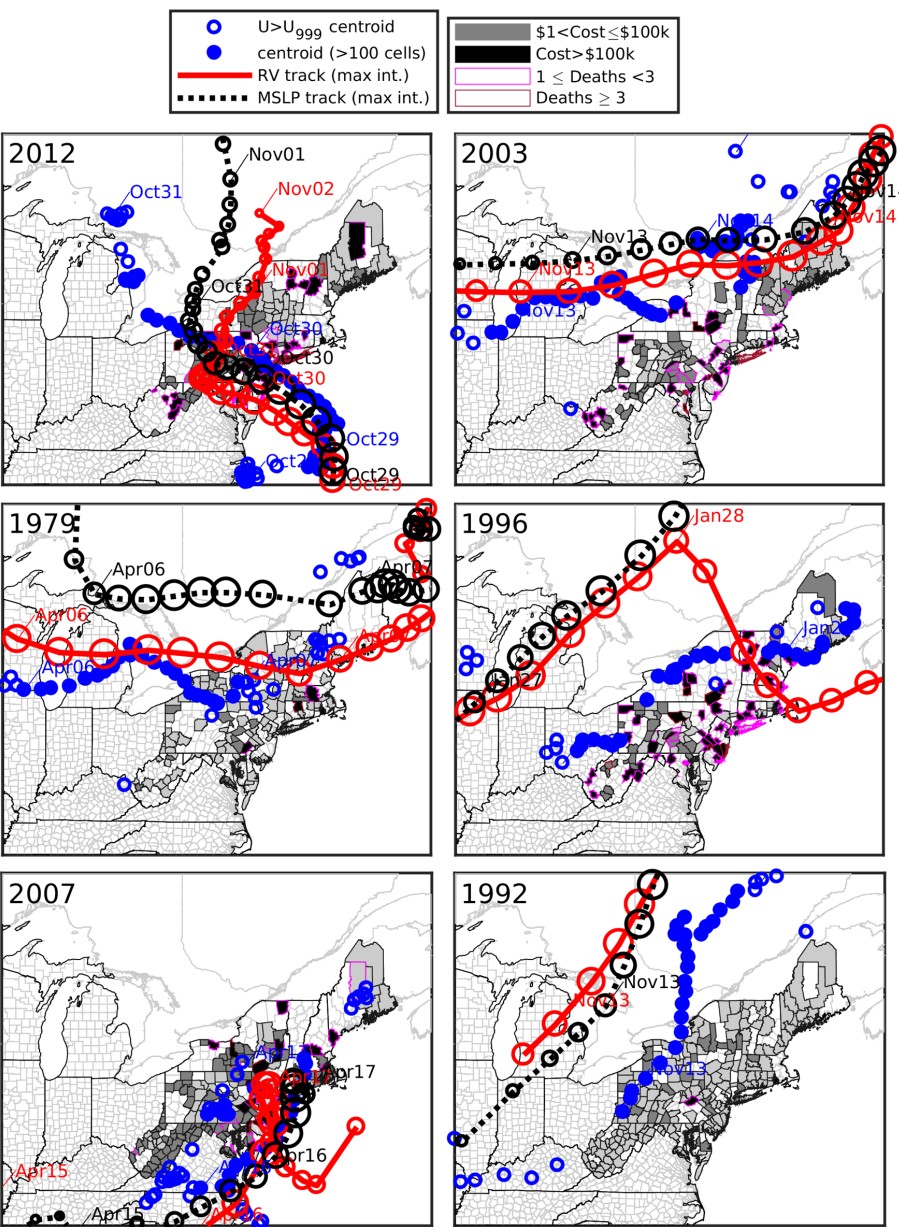

**Figure 3.** TS2

and inflation-adjusted property damage is 0.56. For a sample size of 10, in a t test used to evaluate significance (Wilks, 2011a), these correlation coefficients differ from 0 at confidence levels of 95 % and 90 %, respectively. Excluding Hurricane Sandy increases $r$ between the number of grid cells with $U > U_{999}$ at $t_p$ and inflation-adjusted property damage to 0.86. Thus, this geophysical intensity metric captures aspects of relevance to storm damage.

In previous work, the local 98th-percentile value has been used to identify windstorms in Germany as it roughly corresponds to wind gusts at 10 m that may cause property damage (Klawa and Ulbrich, 2003). Events with widespread ex-

ceedance of the 98th-percentile threshold are common over the US Northeast during the 40 years of ERA5 output. For example, 139 events have sustained wind speeds in excess of their local 98th percentile in over half of all ERA5 grid cells. Thus, herein, a higher threshold (99.9th percentile) is used to distinguish 10 extraordinary windstorms. All 10 also appear on the list of storms chosen using a 98th-percentile threshold, with 9 of the 10 appearing in the top 50 (Table 2).

Several of the windstorms identified using our approach have been previously identified in independent analyses, further confirming the reliability of the detection method. For example, Hurricane Sandy, the most intense windstorm in

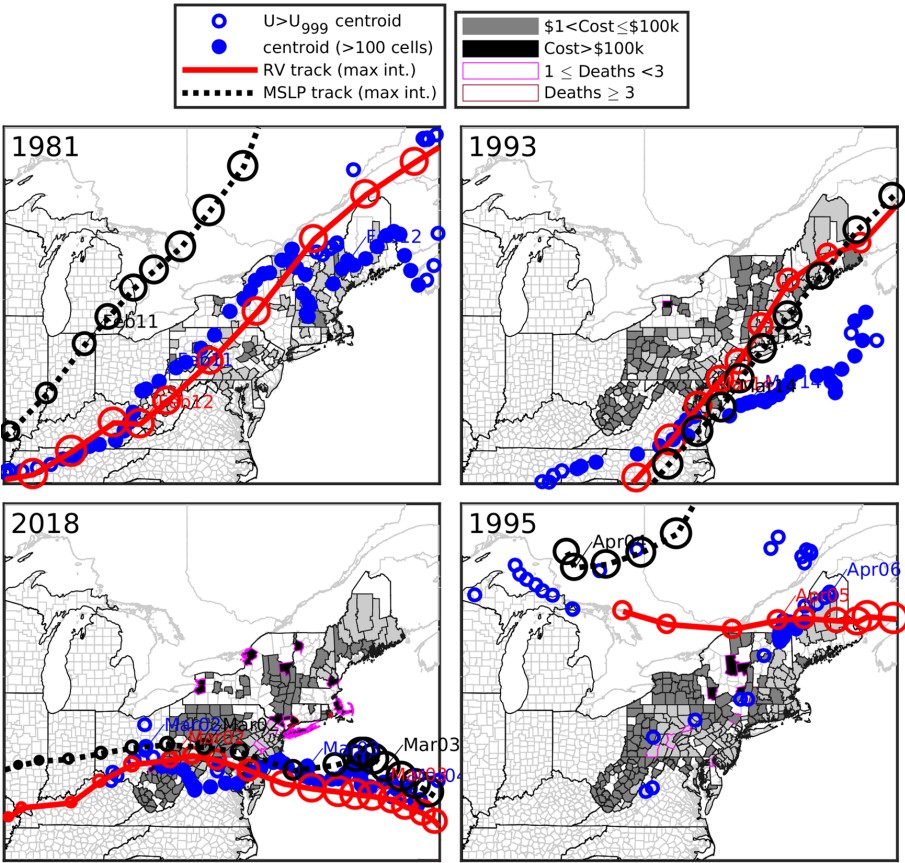

**Figure 3.** Centroids of the windstorms computed as the geographic center of all ERA5 grid cells for $U > U_{999}$ (blue). Markers are filled when there are $>100$ cells over this threshold. Timing and location of the cyclone centers as diagnosed from MSLP and relative vorticity at 700 hPa are shown in black and red, respectively. Markers every 3 h along each track have a diameter corresponding to track intensity. The underlying shading shows the county-level damage and deaths in the northeastern states associated with each event as diagnosed from the NOAA storm reports.

this analysis (Table 2), is a historic storm that moved parallel to the coast before making landfall in southern New Jersey on 29 October and caused USD 50 billion of damage (Lackmann, 2015). According to ERA5 output at its peak, over 300 000 km$^2$ of the northeastern states exhibited wind speeds at 100 m a.g.l. that exceeded the locally determined $U_{999}$ (Fig. 4). The eighth-most-intense windstorm (Table 2) is the "Storm of the Century" of 12–14 March 1993, which formed in the Gulf of Mexico and caused widespread damage in Florida and along the Atlantic coast before entering the Northeast (Huo et al., 1995).

The synoptic-scale structure of extra-tropical cyclones is complicated (Hoskins, 1990; Earl et al., 2017). Maximum wind speeds are often, but not always, associated with low-level jets that occur along the cold fronts of extra-tropical cyclones (Hoskins, 1990; Browning, 2004). Consistent with that expectation, the centroid of ERA5 grid cells with $U > U_{999}$ tends to move in parallel with the cyclone track locations but is generally displaced to the south/southeast (Fig. 3).

Previous research has reported that reinsurance contracts commonly employ a 72 h window to describe a "single event" (Haylock, 2011). All of the windstorms identified in this work transited the northeastern study domain in $<72$ h. Intense wind coverage ($U > U_{999}$) is generally concentrated in the $\pm10$ h around the storm peak time, $t_p$ (Fig. 4), although some windstorms had longer duration and a slower decay in widespread intense wind speeds with significant coverage remaining $>10$ h after $t_p$ (Fig. 4).

Twenty-four-hour precipitation totals, used as an indicator of flooding potential, and maximum precipitation rates, used as an indicator of transportation hazards, vary substantially among the 10 windstorms, but virtually all of the windstorms were associated with some form of intense or hazardous precipitation (Fig. 5). Consistent with observational evidence (Munsell and Zhang, 2014), Hurricane Sandy (windstorm during 2012) is associated with total 24 h precipitation accumulation exceeding 100 mm in five grid cells within the Northeast, and nearly half (46 %) of grid cells exhibit precipitation accumulations of over 20 mm. Heavy precipita-

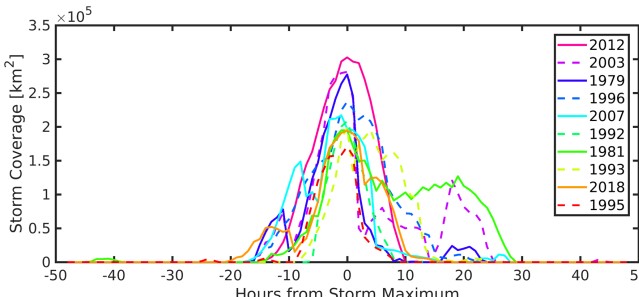

**Figure 4.** Spatial extent of the windstorms measured in square kilometers over the northeastern states using a time index relative to $t_p$. The spatial extent is described as the area of ERA5 grid cells wherein the $U > U_{999}$. Values are shown for 48 h preceding and following each windstorm peak.

tion, both in terms of maximum precipitation intensity and total accumulated precipitation, is also associated with the 1993 windstorm resulting from a decaying TC that formed a NE (Fig. 5). Windstorms with lowest precipitation totals occurred in 2003, 1979, and 1995 and are associated with ACs. Freezing rain, which in conjunction with high winds is a particular hazard to electrical infrastructure and transportation, is present during the windstorms in 1992, 1981, and 1993 (Fig. 5). There is also snow indicated in at least one location in the domain in every storm, except for Hurricane Sandy. Thus, 6 of the 10 windstorms might be classified as compound events due to the occurrence of freezing rain and/or widespread heavy rain identified using the American Meteorological Society threshold of $> 0.76 \, \mathrm{mm \, h^{-1}}$ (AMS, 2012) in $> 40\%$ of grid cells which also exceed $U_{999}$.

Four of the top 10 windstorms occurred after 2000 (2012, 2003, 2007, and 2018, Table 2), and thus high-quality ASOS and radar data are available for comparison with estimates from ERA5 for these events. For the 2012, 2003, and 2018 windstorms there is good agreement between the spatial extent of locally extreme wind speeds from ERA5 and ASOS, and the duration of intense wind speeds (Fig. 6). The agreement is less good for the 2007 windstorm possibly due to the low density of ASOS stations in the US state of Maine, where the ERA5 output indicates the wind maximum was manifest for a substantial fraction of the storm period (Fig. 3). For the other three windstorms the fraction of ERA5 grid cells in the northeastern states with $U > U_{999}$ closely matches the fraction of ASOS stations in the same area that exceed their local $U_{999}$ threshold during each hour of the storm period (Fig. 6). The timing of storm precipitation in the ERA5 data is also in good agreement with observational estimates from radar and ASOS stations, consistent with assimilation of radar precipitation and in situ station data (Lopez, 2011; Hersbach et al., 2019). The period with the most intense precipitation occurred concurrently with the high wind speeds during Hurricane Sandy but largely well before $t_p$ in the 2007 and 2018 windstorms (Fig. 6), consistent with previous work char-

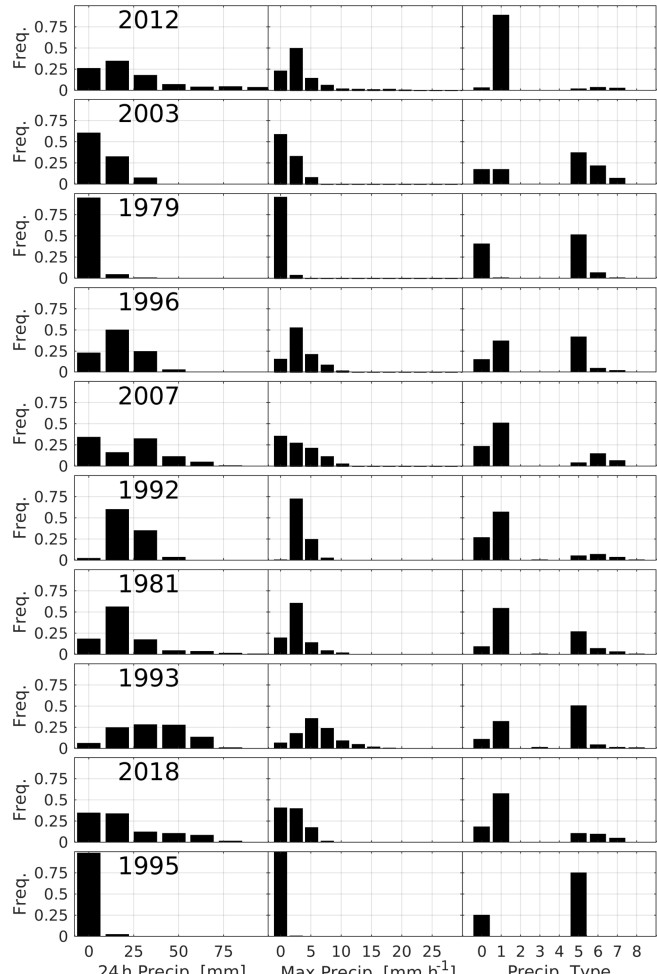

**Figure 5.** Histograms of precipitation totals and maximum precipitation rates and precipitation types for the 24 h centered on each storm peak. All ERA5 land-based grid cells in the northeastern states which exceed their local $U_{999}$ value at any point in the 24 h period are included. The frequencies are the fraction of such grid cells in each class. Precipitation types are as follows: no precipitation (0), rain (1), thunderstorm (2), freezing rain (3), snow (5), wet snow (6), mixture of rain and snow (7), and ice pellets (8).

acterizing extra-tropical cyclones (Bengtsson et al., 2009). Mean ERA5 precipitation rates in Northeast states during these 10 storms are consistently somewhat higher than estimates from radar but below ASOS point measurements, reflecting spatial variability in rainfall intensity at scales below those manifest in a network of point measurements (Villarini et al., 2008).

A larger sample of 50 windstorms is also drawn from the 40-year time series to examine the serial dependence. In this analysis the 14 d exclusion window used in the identification of the top 10 windstorms is reduced to a 2 d window. While the top 10 windstorms considered in detail herein all have a spatial extent of between 309 and 524 grid cells, the storms ranked 11th through 50th in the set used to characterize se-

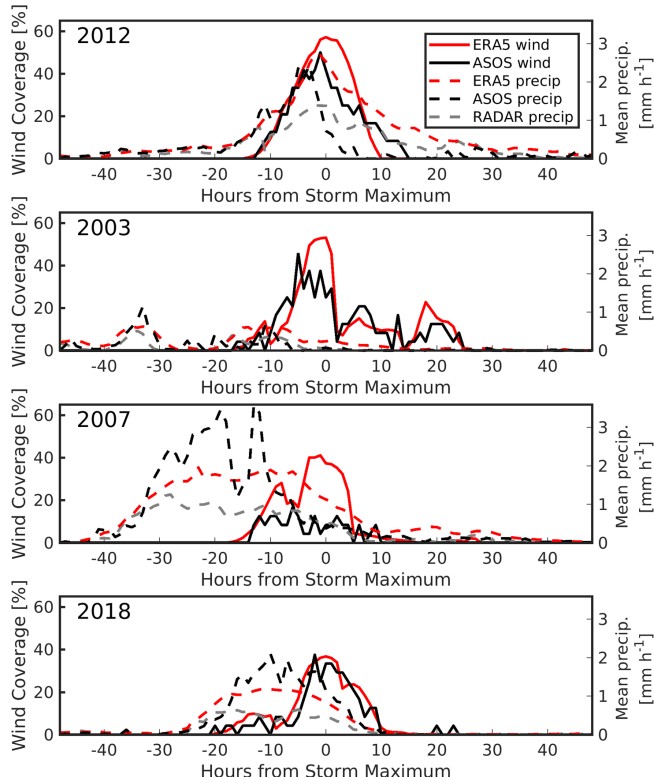

**Figure 6.** Time series of high wind coverage and mean precipitation rate during the four windstorms that occurred after the year 2000. Each subplot includes the fraction of ERA5 grid cells with over-threshold wind speeds ($U > U_{999}$); the number of ASOS stations with over-threshold wind speeds; and the mean precipitation rate (in land areas of Northeast states within 200 km of a radar station) from ERA5, NWS radar, and ASOS point observations.

riality have a mean extent of 216 grid cells and range in extent from 176 to 309 cells, further indicating that the top 10 storms are distinct in the 40-year time series (Fig. 1). One windstorm (on 19 January 1996) is excluded by use of a 14 d separation window from the list of the top 10 storms but is included if a 2 d exclusion period is used. It would have been ranked number 10.

The probability distribution of the annual counts of windstorms is relatively well described by a Poisson distribution. The resulting dispersion value (D) is 0.18, indicating evidence for serial dependence or, alternatively stated, that these windstorms are clustered in fewer years than would be expected for independent events. Of 10 000 bootstrapped samples, 99.97 % had dispersion indices above zero. While this D value (0.18) is symptomatic of serial clustering for windstorms that impact the northern USA, much higher serial clustering was reported for regions of Europe in earlier research using the 20th-century ERA reanalysis and a 98th-percentile wind speed threshold (Walz et al., 2018). The lower amount of serial clustering of windstorms in the northeastern states at the annual timescale is indicative of a lower

probability of multiple damaging windstorm events occurring within a single year.

## 3.2 Cyclone detection and tracking

Consistent with past research employing other reanalysis datasets (Ulbrich et al., 2009), results from application of the cyclone detection and tracking algorithm to ERA5 output also indicate the US Northeast exhibits a high frequency of transitory cyclones (Fig. 7). Also in accord with expectations, the tracks followed by the top 10 windstorms are generally characteristic of those dominant cyclone tracks and derive from a mixture of intense NEs, ACs, CLs, and decaying TCs (Table 3, Fig. 7).

For most cyclones independent tracking of the center using MSLP and RV yields results that are highly consistent (Fig. 3). Nevertheless, some discrepancies exist. These likely arise, at least in part, due to the spectral field smoothing. Another possibility is that, if there is a strong background flow due to a strong pressure gradient, the vorticity can be offset relative to the pressure minimum (Sinclair, 1994).

Cyclone intensities for the top 10 windstorms are an order of magnitude above the mean intensities for cold-weather cyclones at the same locations over the USA for both RV and MSLP (Fig. 8). The median intensity of RV tracks for the 10 storms is $7 \times 10^{-5}$ s$^{-1}$ as compared to $6 \times 10^{-4}$ s$^{-1}$ for all cold-season tracks affecting the Northeast. The median intensity of MSLP tracks for the 10 storms is 25 hPa as compared to 1.2 hPa for all cold-season Northeast storms (Fig. 7, Table 3). Both the 2012 and the 1993 windstorms (ranked no. 1 and no. 8, respectively; see Table 2) are the result of decaying tropical cyclones, with the 1993 system transitioning to become a NE (Figs. 3 and 7, Table 3). The 2012 windstorm exhibited extremely high intensity and is also associated with the largest area (number of grid cells) with $U > U_{999}$. It was also associated with by far the largest amount of property damage and deaths (Fig. 3, Table 2). Five of the 10 storms are associated with Colorado lows, consistent with the high prevalence of such cyclones (Booth et al., 2015) (Fig. 7). These storms generally impacted the smallest areas and tend to be associated with substantial but lower amounts of property damage than TCs or ACs (Table 2).

The 2003, 1979, and 1995 windstorms are associated with Alberta clippers (Table 3) that exhibit initially low intensities but rapidly intensify as they pass across the Great Lakes region ($\sim 45°$ N, 80° W). Cyclone intensities for these three storms increased by an average of 16 % for RV and 33 % for MSLP during their crossing of the Great Lakes longitudes (92 to 76° W). Consistent with a priori expectations, these windstorms occurred when Great Lakes ice cover was minimal (https://www.glerl.noaa.gov/data/ice/atlas/ice_duration/duration.html, last access: 2 June 2021). Both 2003 and 1979 windstorms (ranked no. 2 and no. 3, respectively) exhibit large spatial scales (Fig. 4) and resulted in substantial property damage (Table 2).

**Table 3.** Windstorm details (windstorms are ordered as in Table 2). Cyclone type is based on subjective evaluation of results from the cyclone detection and tracking algorithm. AC – Alberta clipper. TC – tropical cyclone. CL – Colorado low. NE – nor'easter. Max intensity is the maximum cyclone intensity along the storm-associated cyclone tracks for RV ($\times 10^{-5}$ s$^{-1}$) and MSLP (scaled by $-1$, hPa). No. cells with $U_{max}$ indicates the number of grid cells for which the maximum wind speed for the storm year occurred within the storm period. Median RP is the 50th-percentile return period for maximum wind speed in each northeastern grid cell during each storm period, while p95 is the 95th-percentile RP. Also shown is the median RP for grid cells that exhibited $U > U_{999}$ at the storm peak. All RP values include a 95 % confidence interval in parentheses.

| Cyclone type | Cyclone track start | | | Cyclone track end | | | Max intensity: RV [$10^{-5}$ s$^{-1}$] / MSLP [$-1$ hPa] | No. cells with $U_{max}$ | Median RP [years] (95 % CI) | $p_{95}$ RP [years] (95 % CI) | Median RP of cells exceeding $U_{999}$ [years] (95 % CI) |
|---|---|---|---|---|---|---|---|---|---|---|---|
| | Time | Lat [° N] | Long [° W] | Time | Lat [° N] | Long [° W] | | | | | |
| TC | 2012/10/18 09:00 | 11.61 | 61.1 | 2012/11/2 00:00 | 46.92 | 74.95 | 14.3/49.1 | 530 | 4.6 (2.9–9.3) | 105.8 (29.7–583) | 12.2 (5.8–34.8) |
| AC | 2003/11/11 00:00 | 52.97 | 129.82 | 2003/11/23 06:00 | 50.39 | 68.5 | 10.5/36.9 | 494 | 2.3 (1.8–3.6) | 34.9 (12.9–138.3) | 5.5 (3.3–12.1) |
| AC | 1979/4/4 00:00 | 50.61 | 105.62 | 1979/4/8 21:00 | 46.98 | 63.88 | 10.0/32.1 | 412 | 1.6 (1.4–2) | 43.6 (15.6–178.9) | 6.4 (3.7–14.6) |
| CL | 1996/1/26 00:00 | 37.91 | 105.01 | 1996/2/1 06:00 | 57.08 | 41.55 | 10.5/45.4 | 488 | 3.5 (2.4–6.7) | 19.4 (8.3–62.7) | 5.1 (3.1–10.9) |
| CL/NE | 2007/4/11 21:00 | 36.44 | 118.73 | 2007/4/17 18:00 | 39.56 | 69.32 | 12.4/39.6 | 462 | 1.6 (1.4–2.1) | 18.1 (7.9–59.3) | 3.7 (2.5–7.3) |
| CL | 1992/11/12 21:00 | 42.71 | 86 | 1992/11/15 12:00 | 57.06 | 45.63 | 11.2/50.1 | 343 | 1.5 (1.3–1.8) | 6.5 (3.7–14.8) | 3 (2.1–5.4) |
| CL | 1981/2/11 00:00 | 37.44 | 94.5 | 1981/2/16 06:00 | 63.41 | 37.65 | 8.9/56.3 | 523 | 2.2 (1.7–3.4) | 22.2 (9.4–72.1) | 6.6 (3.7–15.1) |
| TC/NE | 1993/3/12 06:00 | 27.37 | 101.4 | 1993/3/15 18:00 | 51.88 | 52.39 | 15.3/49.2 | 536 | 2.1 (1.7–3.2) | 36.8 (13.6–144.1) | 5.4 (3.2–12) |
| CL | 2018/3/1 03:00 | 38.14 | 93.72 | 2018/3/6 06:00 | 42.13 | 53.31 | 13.3/40.9 | 310 | 1 (1–1) | 14.1 (6.5–43.5) | 4.9 (3–10.5) |
| AC | 1995/4/4 15:00 | 45.88 | 80.74 | 1995/4/10 06:00 | 62.63 | 58.16 | 9.5/24.2 | 94 | 1 (1–1) | 14.4 (6.7–42.4) | 2.3 (1.8–3.6) |

Tracking of windstorms is a key determinant of societal impacts. The 2018 windstorm is associated with a CL that stalled over the Atlantic coast and re-intensified to form a NE. Although this event is not the most geographically expansive, its track over very high density population areas and high value assets led to high associated storm damage (Fig. 3). The 2012 and 2018 windstorms passed over highly populated areas, including New York, and are associated with recorded damage in the hundreds of millions of dollars (Fig. 3, Table 2). Conversely, the 1993 windstorm high-wind-speed centroid is out over the Atlantic Ocean, which may partly explain the lower loss of life and property damage associated with this event (Fig. 3). The AC-associated windstorms (2003, 1979, and 1995) track west–east and have maximum intensity centers across the north of the region. They are thus associated with lower damages over the USA than the other windstorms. Cyclones associated with the windstorms in 1992, 1996, and 1981 tracked

from the southeast to the northwest, but their centers diagnosed from MSLP remain east of the region, as do those from RV in 1992 and 1996. The geographic centroids of high wind speeds track through Virginia, Pennsylvania, and New York in all three events, but the advection velocity of the cyclones and the point in their life cycle vary (Fig. 7). Accordingly, inflation-adjusted damage amounts range from USD 24 million for the 1981 windstorm to USD 2181 million for the 1996 windstorm (Figs. 3 and 7).

## 3.3 Windstorm return periods

All 10 windstorms are associated with long-return-period (RP > 50 years) wind speeds in at least some ERA5 grid cells. Data from some grid cells within the Northeast indicate return periods of over 100 years for the 2012 windstorm. Defining a single return period for each windstorm is difficult due to the multiple degrees of freedoms, but the median (50th percentile) and highest 5 % (95th percentile) of ERA5 grid

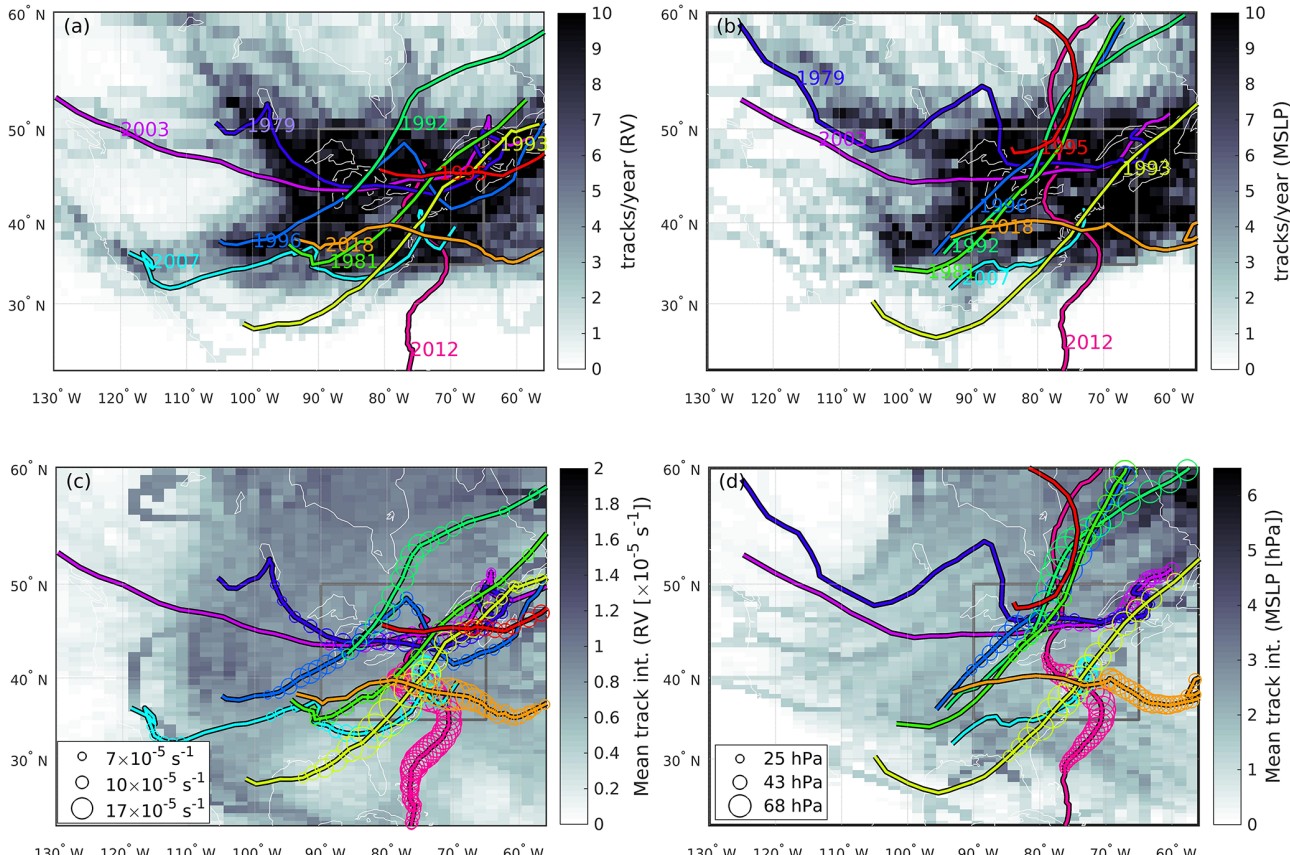

**Figure 7.** Cyclone tracks associated with each of the top 10 windstorms (individual colors) plotted over a heat map of cyclone densities for **(a)** relative vorticity (RV) and **(b)** mean sea level pressure (MSLP). Background cyclone densities and intensities include only cold-season storm tracks that enter the Northeast rectangle. Cyclone intensities for analyses of **(c)** 700 hPa RV and **(d)** MSLP (shown as an absolute value) for each of the top 10 windstorms (where the symbol diameter scales with intensity) plotted over a heat map of mean cyclone intensities. Symbol sizes shown in the figure legends represent the 50th-, 70th-, and 90th-percentile cyclone intensities from among the top 10 windstorms. Tracks have no intensity markers when they are below the 50th-percentile intensity. Track densities and intensities in all four panels are computed at the ERA5 grid resolution and then averaged to a $1° × 1°$ grid to aid legibility. These background field values include only cyclones that track into the Northeast rectangle (shown in grey) during cold months (October–April 1979–2018) and are anomalies identified in the filtered fields, obtained from the spectral filtering which has the large-scale background removed for the tracking. Color coding of the cyclone tracks associated with each windstorm is as in Fig. 4.

cell estimates provide some qualitative assessment of probability. The median RP computed for all 924 grid cells ranges from 1 to 5 years across the 10 windstorms (Table 3), while at least 5 % of grid cells are characterized by wind speeds during each of the 10 windstorms with RP of 6.5 to 106 years (Table 3, Fig. 8). The number of ERA5 grid cells that exhibit their annual maximum value during the storm period is positively correlated with the three metrics of return periods: (i) median RP, (ii) 95th-percentile RP, and (iii) median RP for grid cells that exhibited $U > U_{999}$ (*r*: 0.45 to 0.64), consistent with the longest-RP wind speeds being associated with the largest windstorms (Fig. 8, Table 3). For the two windstorms caused by TCs that entered the northeastern states (2012 and 1993), high RP wind speeds are concentrated along the coast. The 2003 and 1979 windstorms, the highest-magnitude Alberta clippers, are associated with ex-

tremely high return period wind speeds in the Great Lakes region. Wind speeds over a large number of grid cells over and around the Great Lakes had a RP of >50 years during the 1979 windstorm. Indeed, this windstorm, while not the most spatially expansive (Table 2), is the event with the largest number of ERA5 grid cells in excess of 50-year RP wind speeds in the Northeast domain. The Colorado-low-associated windstorms (1996, 2007, and 1981) have their highest RP winds in the mountainous regions of West Virginia, New York, Vermont, and Maine (WV, NY, VT, and ME, respectively).

Extrapolation to low-probability, long-return-period wind speeds from limited-duration time series is naturally associated with substantial uncertainties (Wilks, 2011b). For example, the 95 % confidence intervals on the 95th percentile of grid cell RP values during the 10 windstorms range from 30

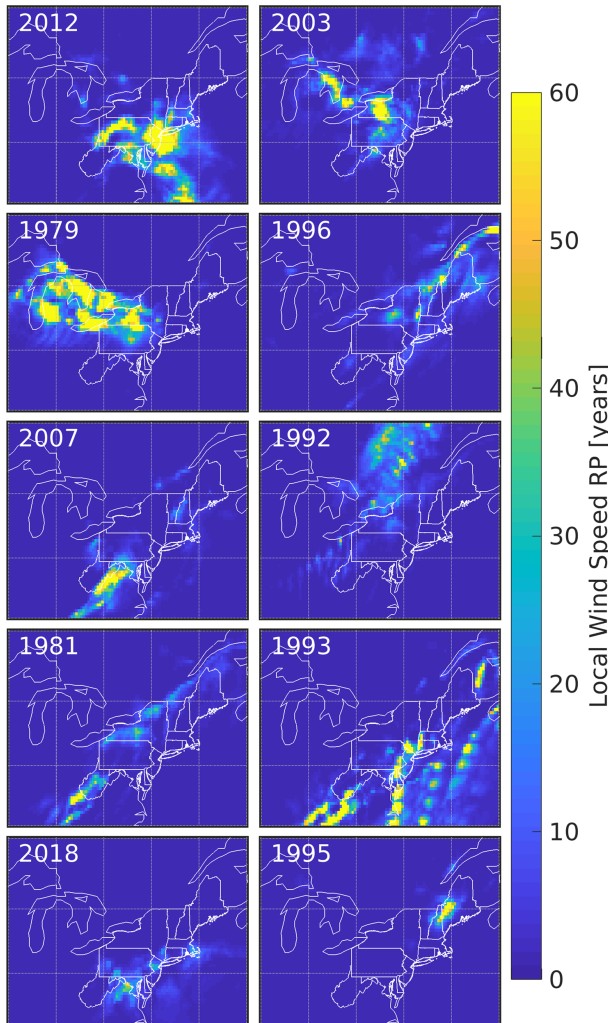

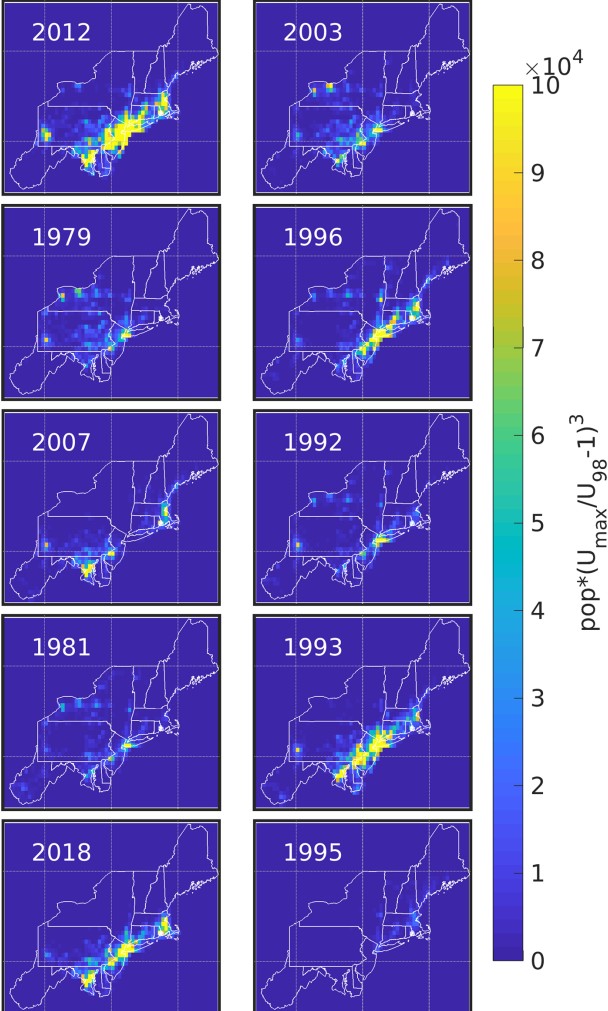

**Figure 8.** Return period (in years) of storm-maximum wind speed ($U_{peak}$) in each ERA5 grid cell associated with each windstorm. The color scale is truncated at 60 years for legibility. But, for example, the RP of the maximum wind speeds at 100 m during Hurricane Sandy (2012) exceeds 100 years for multiple grid cells. Northeastern state borders and coastlines (Atlantic Ocean and Great Lakes) are shown in white.

**Figure 9.** Contribution to the loss index (LI; Eq. 4) from each ERA5 grid cell associated with each windstorm. Northeastern state borders and coastlines (Atlantic Ocean and Great Lakes) are shown in white.

to over 500 years for Hurricane Sandy with a best estimate of 106 years (Table 3). Irrespective of the precise RP for these windstorms, this analysis emphasizes the truly exceptional nature of these events.

### 3.4 Loss indices and comparison to NOAA storm damage estimates

Population weighting mean loss index contributions (Eq. 4) for the 10 windstorms identified herein are generally maximized in the coastal grid cells that comprise the northeastern urban megapolis that extends from New Jersey to Massachusetts and includes the city of New York (Fig. 9).

Both LI and the number of ERA5 grid cells in NE states exceeding their 99.9th-percentile wind speed exhibit positive correlations with the NOAA storm damage report totals for the windstorms. A linear fit with zero intercept of NOAA storm damage in millions of USD (inflation adjusted to January 2020) and the number of cells exceeding $U_{999}$ exhibits variance explanation ($R^2$) of 0.24 and a slope of $1.1 \times 10^7$. A linear fit of NOAA storm damage and the LI has an $R^2$ of 0.75 and a slope of 554. A substantial fraction of variability in economic losses associated with these 10 very high magnitude and large-spatial-extent windstorms is not well described solely by the number of grid cells with $U > U_{999}$ at $t_p$. This is partly due to co-occurrence of other geophysical hazards (including flooding due the composite nature of some of these events; see Fig. 5). For example, the 2012 storm (Hurricane Sandy, ranked no. 1 in this analysis) is associated with greater property damage than would be predicted by either

the LI or number of cells exceeding $U_{999}$, due to damage from storm surge and related flooding (Xian et al., 2015). Excluding Hurricane Sandy, the $R^2$ value computed using Eq. (5) for a linear fit with zero intercept between the NOAA storm damage and the number of cells exceeding $U_{999}$ decreases to 0.13, and that for the relationship between LI and NOAA storm damage decreases to 0.16. This is partly because population density is a crude index of socioeconomic exposure or the presence of high-value assets. Future work could explore the degree to which inclusion of a wealth index improves these associations (Pielke and Landsea, 1998).

## 4    Concluding remarks

The US Northeast exhibits high socioeconomic exposure to atmospheric hazards due to the presence of major urban centers with high population density and high density of insured, high-value assets (Table 1, Fig. 1), and windstorms present a substantial fraction of historically important climate hazards in this region. The northeastern states are also experiencing population increases that are projected to continue into the future (Zoraghein and O'Neill, 2020). This increase in population may result in increased exposure to this hazard even in the absence of any change in windstorm frequency or intensity. Thus, there is great value in improved characterization of these events.

The 10 most intense windstorms in the northeastern USA during 1979–2018 covered 33 % to 57 % of ERA5 land cells in the northeastern states with wind speeds exceeding the locally determined 99.9th-percentile threshold (Table 2). Although all 10 events occurred during the cool-season months of October through April, they are distributed throughout the 40 years, and no individual year exhibits more than one of these events (Fig. 1b). However, when a larger pool of the top 50 largest windstorms is considered, evidence of serial clustering emerges. Return periods for wind speeds in the upper 5 % of ERA5 grid cells during these 10 windstorms range from 6.5 to 106 years (Table 3, Fig. 8). Many of these windstorms exhibit co-occurrence of extreme and/or hazardous precipitation and thus may be considered composite events.

Any windstorm catalogue is, to some degree, a product of the dataset on which it is predicated, and the windstorms identified herein are derived using a methodology that preferences intense but large-scale events. Their characteristics will naturally differ from severe local storms. The windstorms identified independently and objectively in this work are consistent with historically notable events. Further, precipitation and wind speeds from ERA5 for windstorms that occurred after 2000 exhibit good agreement with in situ observations from the NWS ASOS network and NWS dual-polarization radar, consistent with assimilation radar precipitation and weather station data streams by the ECMWF data assimilation protocols and past evaluations of the ERA5 reanalysis (Fig. 6). The statistically significant correlation between the LI or number of cells exceeding $U_{999}$, due to damage ERA5 windstorm intensity estimates and independent damage estimates provides further confidence in the fidelity of the windstorm catalogue presented herein.

The cyclone tracks associated with the 10 windstorms are consistent with the climatology of cold-season cyclones, and thus the associated extra-tropical cyclones are a mixture of Alberta clippers, Colorado lows, decaying tropical cyclones, and nor'easters (Fig. 7). These cyclones, however, exhibit intensities (from both RV and MSLP perturbations) that are an order of magnitude higher than mean values sampled on those same tracks (Fig. 7). With the possible exception of Hurricane Sandy, these windstorms follow tracks that are not infrequent in the cyclone climatology. It is also notable that the most intense AC events occurred during periods of low ice cover in the Great Lakes, which may imply windstorms associated with AC events are likely to intensify under climate change as a result of reduced icing of these water bodies (Smith, 1991).

Inflation-adjusted (to January 2020) property damage totals for each of the windstorms range from USD 24 million to USD 29 billion (Table 2). While there is not perfect agreement in the ranking of these storms between high wind coverage and property damage, the top four storms in terms of extent do all have higher damage totals than the next six.

This windstorm catalogue is intended to characterize extreme windstorms in the northeastern USA and may have value in efforts to evaluate and validate climate and natural hazard catastrophe models. Planned extension of the ERA5 reanalysis to 1950 may provide an opportunity to further extend this analysis to include elements related to non-stationarity in windstorm probability, with the caveat that such detection will be challenging due to changes in the assimilated data. Research is underway to dynamically downscale these windstorms using the Weather Research and Forecasting model to examine sub-grid-scale variability in extreme wind speeds and the sensitivity of these events to global climate non-stationarity.

*Data availability.* ERA5 reanalysis output are available from https://climate.copernicus.eu/climate-reanalysis (Copernicus, 2021). NWS radar data are available from the National Climatic Data Center: https://www.ncdc.noaa.gov/data-access/radar-data (NCDC, 2021). NWS ASOS data are available from ftp://ftp.ncdc.noaa.gov/pub/data/asos-fivemin/ (NCEI, 2021a). The NOAA Storm Events Database is available at https://www.ncdc.noaa.gov/stormevents/ (NCEI, 2021b). Historical estimates of Great Lakes ice cover are available from https://www.glerl.noaa.gov/data/ice/atlas/ice_duration/duration.html (NOAA, 2021).

*Author contributions.* All four authors participated in discussions about the goals and methods for this paper. SCP devised the analysis framework. FL had primary responsibility for performing the analyses. FL, SCP, and RJB wrote the majority of the manuscript

text. KIH provided analysis tools, expertise, advice, and context for cyclone tracking. RJB and SCP performed analyses on the societal impact of these windstorms. RJB and SCP acquired the funding and computing resources to make this research possible.

*Competing interests.* The authors declare that they have no conflict of interest.

*Acknowledgements.* This work was made possible by computing resources from the National Science Foundation: Extreme Science and Engineering Discovery Environment (XSEDE) (allocation award to SCP is TG-ATM170024). The thoughtful comments from the anonymous referees improved the clarity of the manuscript.

*Financial support.* This research has been supported by the U.S. Department of Energy (grant nos. DE-SC0016438 and DE-SC0016605).

*Review statement.* This paper was edited by Joaquim G. Pinto and reviewed by two anonymous referees.

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

**Remarks from the typesetter**

TS1 Due to the requested changes with influence on the content of this table, we have to forward your requests to the handling editor for approval. To explain the corrections needed to the editor, please send me the reason why these corrections are necessary. Please note that the status of your paper will be changed to "Post-review adjustments" until the editor has made their decision. We will keep you informed via email.

TS2 Please note that in line with our house standards, the figure caption is inserted only for the last part of the figure.

TS3 Please confirm that the dates were formatted correctly in column 2 and 5.