# Peer review of "Intense windstorms in the Northeastern United States"

_Natural Hazards and Earth System Sciences, 2020_

## Referee Comment (RC1) · Anonymous Referee #1 · 7 Jan 2021

This study presents an identification of severe windstorms over northeastern North America. Objective cyclone identification and wind speed exceedance methods are used to identify these phenomena and their dynamical features and impacts are put into the context of the wider climatology. The aims of the paper are good and I believe it could make an interesting publication that is appropriate for the journal and would provide a good piece of analysis to partner the numerous similar studies over Europe. However, I believe many of the methods used are inconsistent with these other studies and are not sufficiently justified or argued for in the text. All the methods used are slightly different from those established in the literature, such as tracking at 700 hPa instead of 850 hPa, analysing wind gusts at 100 metres instead of 10 metres, and using an extremely high exceedance threshold of the 99.9th percentile instead of the

more commonly used 98th percentile. I would like to see some evidence for the choice of the methods. Furthermore, the introduction and framing of the paper (especially the first half) feels very incoherent and I believe requires significant re-structuring. In addition, the presentation of the figures feels clumsy and very difficult to interpret with numerous similarly styled lines and colours overlayed on similarly shaded fields, which needs rectifying. My full points can be found below. I therefore recommend that this study requires major corrections.

Individual points

1. L43 – I feel the reference here to Fig. 1a should illustrate the two different types of cyclone tracks you are describing. As the figure is for wind gusts it in no way illustrates the differences in the genesis location

2. L47-49. How is the influence of post-tropical cyclones in the USA consistent with Europe? Are less than 1% of cyclones also post-tropical in this region? Perhaps it is best to remove the 1% statement.

3. L61-63. Can a trend line or some evidence of the trend from the data in Fig. 1b also be included/quoted? Due to the large inter-annual variability of the data it is hard to tell from the figure you have presented that there is a positive trend. Furthermore, you quote the 98th percentile in the text (perhaps this refers to the results of Bronnimann et al., 2012), yet the figure in question (I assume Fig. 1b?) uses number of grid points exceeding the 99.9th percentile of U. Please clarify these differences, or make the figure consistent with the text.

4. Section 1.3. From your introduction there is limited evidence of where this study fits within the established research and scientific literature. I think it could be framed better to give a clear narrative as to where the gaps in the literature are and how this study addresses those gaps.

5. L123-124. I feel this statement is incorrect for the last 10-15 years with the in-

troduction of more advanced and homogenous reanalysis products. Please make it clearer how ERA5 is beneficial compared to previous generation products such as ERA-Interim, etc.

6. I am a little confused as to why the 100 metre level is being used throughout this study as you are specifically interested in damaging windstorms. You have justified this in the text, however I feel it may be better to be consistent with previous studies, which generally use the 10 metre level. Does using 100 metres lead to drastically different results to using 10 metres? Furthermore, as you are using near-surface ground observations as a validation, surely it would make sense to also use the 10-metre product from ERA5 as there could be significant differences in the 100m and 10m distributions.

7. As above, doing the tracking at 700 hPa instead of 850 hPa is also confusing to me. Using the Hodges method tracking is mostly done at 850 hPa. Are there specific issues with representing the flow in the orographic regions? Are all cyclones identified at 700 hPa the same as those that would be identified at 850 hPa?

8. I feel the methodology section can be condensed significantly. You introduce the tracking on line 148 and then describe it in detail, several pages later. It feels very incoherent and should be re-structured for conciseness.

9. L210 – you use the 99.9th percentile, what is the justification for this? As you mentioned in the text your method is similar to numerous other storm severity assessments, however most use the 98th percentile. Due to the relative short extent of data from ERA5, are there not chances that the 99.9th percentile is exceptionally skewed by very large events?

10. L223 – is it possible that by using such a high threshold some spatially large events (p98<U<p99.9) are missed and events with a very small area of U>99.9 are counted instead? It may be that in some of these cases could the large, yet slightly lower intensity events have larger impacts than small scale high-wind events.

[Figure]

11. L225 – a 14 day restriction seems rather large. Have you tested this criteria to see if any high impact storms are excluded as a result of this threshold? Several studies have shown that more intense cyclones are more likely to cluster (e.g. Mailier et al., 2006), therefore this could be removing events from your set.

12. L235 – further references are needed here such as Vitolo et al. (2009), Mailier et al. (2006), Pinto et al. (2014).

13. L265 – do you have specific requirements for each of these cyclone classes (i.e. genesis location). If so please state this in the text.

14. Fig 2. I find the layout and organisation of this figure very messy. The legends should be moved outside of the panel boundaries and all text on the panels be made clearer as it is very hard to decipher any of the information.

15. L317 – reference should be outside of brackets.

16. L348-349 – as discussed above. If your systems traverse your region of interest in ∼72 hours, why the 14 day separation? Please clarify this.

17. Fig 4 and throughout – is it worth referring to each storm by its ranking in the text and figures. I find it hard to keep track of which year is which through the text, this may be a simple way to avoid this.

18. L396-406 – would the authors be able to illustrate these results in some way instead of just giving a description. Furthermore, commonly dispersion is calculated as counts per month, or counts per winter season, and not counts per calendar year. Vitolo et al. (2009) Fig. 15 demonstrates how dispersion (and surrounding uncertainty) commonly increases with aggregation period and as these storms you are interested in are only features of approx. 6 months of the year is it likely that this dispersion value is representative?

19. Figs 6 and 7 – legibility of these figures is very difficult. Often green lines are plotted above an area of green (especially figure 7a) and also with text on the figures it

makes it very difficult to distinguish and correctly identify features. I would recommend a redesign of these figures to aid legibility.

20. Fig 7b and table 3 – are the mslp units in hPa the anomaly from the background field? This needs to be made clearer as the magnitude of the values are confusing.

21. L412 – again, do you have requirements for these cyclone classes. It would be useful to define these earlier in the text instead of just approximating by eye if they are one class or the other. This is also applicable for L416 where you state a cyclone transitions to the NE class.

22. L464-465 – please display this information somewhere in either a figure or table, with it being stated as is it feels unjustified. Also you state how all storms have RP >50 years in some location, is this evident in figure 8 as the colorbar does not extend beyond 50? Perhaps to make this clearer the authors should extend the colorbar beyond 50 years and then highlight the regions which exceed 50 years.

23. Figure 8 and table 3 – what is the uncertainty on the return period calculations? As with ERA5 there is only ~40 years of data the uncertainty must be very large for the 100 year event. Please clarify and quantify this in the text and figures.

24. All figures – the visualisation and colour clashes at times makes for figures that are very hard to interpret. Fig. 8 the red lines of states over the map is almost impossible to clearly distinguish. Fig 6 and 7 the track lines an intensity circles are hard to see as they overlap and also clash with the background colours. Figure 2 has legends overlapping figure space and also text on figures that cannot be read. I feel a redesign of these figures is required to accurately present the authors results.

25. The storm severity metrics (following Klawa and Ulbrich, 2003) are introduced but in no way used in the analysis and I feel could be a strong contribution to the results. Performing an SSI-like calculation to compare ERA5 ranking to actual ranking would be a useful addition to the analysis. It would be good to compare the U_999 spatial

extent and maximum wind speed to this storm loss metric.

References

Mailier, P. J., Stephenson, D. B., Ferro, C. A., & Hodges, K. I. (2006). Serial clustering of extratropical cyclones. Monthly weather review, 134(8), 2224-2240.

Pinto, J. G., Gómara, I., Masato, G., Dacre, H. F., Woollings, T., & Caballero, R. (2014). Large‐scale dynamics associated with clustering of extratropical cyclones affecting Western Europe. Journal of Geophysical Research: Atmospheres, 119(24), 13-704.

Vitolo, R., Stephenson, D. B., Cook, I. M., & Mitchell-Wallace, K. (2009). Serial clustering of intense European storms. Meteorologische Zeitschrift, 18(4), 411-424.

---

## Referee Comment (RC2) · Anonymous Referee #2 · 15 Jan 2021

Review of "Windstorms in the Northeastern United States" by Frederick W. Letson, Rebecca J. Barthelmie, Kevin I. Hodges and Sara C. Pryor

The paper presents an analysis of the 10 largest windstorms over the northeastern United States during the past four decades. It combines metrics of extreme winds and precipitation based on both reanalyses and observations with tracks of the associated extratropical cyclones based on both pressure and vorticity. The time evolution of reanalysis and observational data qualitatively agrees during the storm events. The storms show typical tracks for the region but intensity about one magnitude higher than average. Several storms are associated with extreme damages over 1B$ and their winds show long return periods above one century locally.

The regional focus on the northeastern United States complements earlier windstorm

studies that focus on northwestern Europe mostly. In that sense, the paper is an important contribution to the windstorm community. However, it suffers from several shortcomings that limit its actual impact. The paper tend to cover too many topics without a clear focus and to combine too many approaches applied in an unusual way. It would benefit from a well-defined scope, standard methods taken from the literature and a better structure altogether. General and specific comments are listed below to help improve the paper quality.

GENERAL COMMENTS

I. The scope of the paper is vague: it appears to be the 10 most intense (or largest?) windstorms over the 40 year period 1979–2018 but this is not clearly stated or motivated. Ten is not enough for a catalog and not significant to represent severe storms but likely too much for detailed case studies. The authors somehow need to choose between increasing the sample and focusing on a few particular storms.

II. The methods are comprehensive (4 pages of description) but uncommon and not sufficiently explained or motivated: 100 m rather than 10 m winds, 99.9th rather than 98th percentile, intensity defined as instantaneous spatial extent of extreme wind rather than the footprint of, e.g., cubed winds above some threshold. This may result in the obtained ranking of storms not matching their damage.

III. Analyzing compound events is certainly meaningful but throughout the paper it is unclear whether the damages associated with the 10 selected storms are due to wind, precipitation, or both. This is somehow linked to the comment above, which applies to precipitation metrics as well.

SPECIFIC COMMENTS

l. 1 The title should be more specific, e.g., referring to the most severe windstorms in the region

l. 14–15 "Alberta Clippers", "Colorado Lows" and "Nor'easters" sound too specific for

the abstract

l. 19 Why "those windstorms that occurred after the year 2000" only? What about the others?

l. 28 In the midlatitudes?

l. 36–37 The second part of the sentence appears unnecessary

l. 43 What is to be seen on Fig. 1a?

l. 48–49 I do not fully understand the sentence and the cited paper describes Europe only

l. 75–99 (Section 1.2) Many studies and numbers are cited but they lack focus on the topic of the paper: windstorms over the northeastern US

l. 86 Repetition of l. 33–34

l. 101–102 This requires more details here and should somehow appear in the title and abstract

l. 105 Most intense or severe?

l. 117–120 This paragraph is unclear and seems misplaced

l. 120 A description of each Section is expected here

l. 121ff (Section 2) The study period is unclear and inconsistent between subsections

l. 123 Over long periods? It is not an issue for case studies

l. 128–129 Hourly or every 20 minutes?

l. 137–144 The motivation for using winds at 100 m agl is not fully convincing and slightly repetitive; this height is used for wind power mainly and may not reflect the surface impact of windstorms

l. 145 Why 3-hourly rather than hourly as above?

l. 155 Which ten storms?

l. 168–171 I agree but this does not support the use of 100-m winds from ERA5

l. 175 Twice "product(s)"

l. 179–183 The two sentences sound repetitive

l. 193–195 Verb missing?

l. 194 What is the caveat here? The impact is indeed expected to depend on population density

l. 210–212 This approach reflects the severity rather than intensity (storm severity index)

l. 212–215 The motivation for using the 99.9th percentile is unclear; although the approach is disputed by other authors, the choice of the 98th percentile arises from comparison with impact data in Klawa et Ulbrich (2003)

l. 215–217 Why?

l. 225 14 days appear as a very long period to separate events

l. 227 if values are instantaneous then +/-48h makes 96h

l. 235–236 The focus on the 10 most intense (severe?) storms should be stated clearly and early in the paper

l. 244 Reference without brackets

l. 259–266 Do the centroids of extreme ERA5 winds actually coincide with extratropical cyclones, in general and for the 10 most intense cases in particular?

l. 267–268 If I understand correctly the intensity is solely defined as the maximum instantaneous areal extent of extreme winds?

l. 299–305 (Table 2) The windstorm category would be useful here (AC, CL, TC, NE)

l. 306–311 What are the actual most damaging extratropical cyclones for the considered period, and are they captured by the selection?

l. 313–320 An additional but crucial explanation may be that the used metric is simply not appropriate... In addition to different wind height and percentile, standard metrics are based on a footprint and of some function of winds (typically cubed) rather than on an instantaneous spatial extent

l. 334–336 The strongest winds often occur to the south of the pressure center but not necessarily related with the cold front (see, e.g., the Hewson and Neu 2015 paper cited above)

l. 340–345 (Figure 2) What is the reason for the large discrepancy between RV and MSLP tracks in some cases?

l. 350–351 It would be interesting to know why it is the case (1981 and 2003 storms)

l. 362 Hurricane Sandy was already introduced

l. 363–364 How many is "several" or "multiple" grid cells? 20 mm in 24 h is far from extreme...

l. 370 (Figure 4) The histograms do not emphasize extreme (i.e., potentially damaging) precipitation. And what is precipitation type 2?

l. 387–390 The underestimation of station observations may be explained by the spatial variability of precipitation but there is a factor ∼2 between ERA5 and radar estimates

l. 396–406 There is a confusion between clustering on yearly and daily time scales; I would expect the analysis focuses on the latter to distinguish between consecutive storms

l. 407ff (Section 3.2) This Section generally lacks structure

l. 408–412 How are these cyclones selected? Based on thresholds in MSLP or vorticity? The total number of extratropical cyclones likely exceeds 10 tracks per year by far

l. 410 What are transitory cyclones?

l. 416–419 This was already stated above.

l. 425–429 This belongs to the Introduction

l. 436 The impact of the 2018 (not 2017) windstorm may be due to precipitation rather than wind mostly

l. 436–447 The impact is already discussed above (e.g., l. 418–424) and in other sections. Please reorganize

l. 450–460 (Figures 6–7) Same question as l. 340–345 (Figure 2)

l. 463ff (Section 3.3) I am lost in the numerous details and wonder what to learn from this additional section

l. 496–502 This belongs to the Introduction

l. 503 It is the first mention of the "largest" windstorms

l. 518–520 I do not agree on the accord with damage estimates

l. 530–533 This is not convincing

l. 560ff (References) Please . . . increase . . . spacing . . . between . . . lines

---

## Author Comment (AC1) · 18 Feb 2021

Thank you for your comments and suggestions. Please find our response to referee comments and tracked-changes manuscript attached.

Please also note the supplement to this comment:
https://nhess.copernicus.org/preprints/nhess-2020-345/nhess-2020-345-AC1-supplement.pdf

---

## Author Response (AR1)

**Response to both referees for NHESS-2020-345**

**Intense windstorms in the Northeastern United States**

Thank you both for your thoughtful comments on our work. Before going on to address your specific comments, we would like to address our use of (i) a wind speed threshold of the local 99.9[th] percentile (rather than 98%) and (ii) 100-m once-hourly sustained wind speeds rather than 10-m gusts. Regarding point (i) our research is seeking to identify truly exceptional windstorms. Use of a 98[th] percentile value is not very discriminatory (see details below). Regarding point (ii) while ERA5 like many reanalysis systems does generate wind gust estimates, when derived at a grid resolution of approximately. 30 by 30 km they are systematically negatively biased relative to observations. Further 10-m wind speeds are strongly dictated by the roughness lengths used in the reanalysis models. Thus, we feel it is appropriate to use variables that are likely to be more presentative and simulated with higher fidelity. We hope this clarifies our perspectives and elaborates on materials within the submitted document that discuss the latter point. Quoting from the original submission;' However, it is important to acknowledge that wind parameters from any model do not fully reflect all scales of flow variability (Skamarock, 2004) and underestimate extreme wind speeds (Larsén et al., 2012), particularly in areas with high orographic complexity and or varying surface roughness length. Here we use wind speeds at 100-m because flow at this height is less likely to be impacted by sub-grid scale heterogeneity in surface roughness length and uncertainties induced by unresolved sub-grid scale variability. Near-surface wind speeds are strongly coupled to wind speeds at 100-m (i.e., within the PBL) but wind speeds at 100-m are less strongly impacted by inaccuracies and/or uncertainty in surface roughness length ($z_0$) (Minola et al., 2020;Nelli et al., 2020). Applying an uncertainty of a factor of two to $z_0$ can lead to mean differences of up to 0.75 ms$^{-1}$ for near-surface (40 to 150 m a.g.l.) wind speeds (Dörenkämper et al., 2020).'

In response to your comments regarding this matter, we have added new analysis of 10-m gusts and a comparison of our storm-ranking algorithm to one using a 98[th] percentile threshold. Related changes are as follows;

1) In section 2.1 we describe the data; 'Wind gusts at a nominal height of 10 m are generated as a post-processing product from the ERA5 reanalysis product. Wind gusts estimates are derived from the sustained wind speed at 10 m along with a term representing shear stress and a convective term (Molina et al. 2020). They are also presented herein to provide a link to previous research on European windstorms that has focus on wind gusts. The association between these wind gust estimates and sustained wind speeds at 100 m are also presented. '

2) Wind gust values at 10 m for each storm are now briefly summarized in Table 2 and described in the associated text. 'Maximum wind gusts at 10 m a.g.l. exceed the sustained wind speeds at 100 m a.g.l. at both the peak hour and over the entire windstorm. Maximum wind gusts from ERA5 for all windstorms are well above the U.S. National Weather Service 'damaging winds' threshold of 25.7 ms$^{-1}$ 1 (Trapp et al., 2006) (Table 2). The spatial correlation coefficient between 100-m sustained wind speeds and wind gusts at 10 m at $t_p$ is > 0.68 for all storms and > 0.8 for 8 out of the 10 storms, indicating that the 100-m sustained wind speeds analyzed herein are strongly related to near-ground wind gusts in the ERA5 reanalysis.

3) The high degree of association between 10-m wind gusts and 100-m sustained wind speeds is now made explicit (see above.).

For the reviewers' interest (and that of any readers who have sought out this document) in the following figure we show the relationship of 10-m gusts to 100-m hourly winds at the peak time of each windstorm.

[Figure]

While retaining the 99.9[th] percentile wind speed threshold central to this work, we have presented information about use a the 98[th] percentile. See table 2 and the related text; 'In previous work, the local 98[th] percentile value has been used to identify windstorms as it roughly corresponds to wind gusts at 10-m gusts that may cause property damage (Klawa and Ulbrich, 2003). Events with widespread exceedance of the 98[th] percentile threshold are common within the 40 years of ERA5 output. 139 events have sustained wind speeds in excess of their local 98[th] percentile in over half of all ERA5 grid cells over the Northeastern states. Herein, a higher threshold (99.9[th] percentile) is used to distinguish ten extraordinary windstorms. All ten also appear on the list of storms chosen using a 98[th] percentile threshold, with nine of the ten appearing in the top 50 (Table 2).'

At the reviewers request we have also included a Loss Index calculation based on Klawa and Ulbrich's 2003 work. This is described in the methods section (2.7) and results are reported in section 3.4.

The following figure shows the relationship between storm damage and number of cells exceeding $U_{999}$ and Loss Index. We have elected not to include this figure in the manuscript because it only confirms the fact the much of the variance in storm damage is not well represented by either metric. As with the figure

above, we have included this figure in our response for the reviewers' interest and for any future readers who seek out this document.

[Figure]

A tracked-changes version of the manuscript is included at the end of this response.

Responses to specific comments below are in **bold.**

Anonymous Referee #1

This study presents an identification of severe windstorms over northeastern North America. Objective cyclone identification and wind speed exceedance methods are used to identify these phenomena and their dynamical features and impacts are put into the context of the wider climatology. The aims of the paper are good and I believe it could make an interesting publication that is appropriate for the journal and would provide a good piece of analysis to partner the numerous similar studies over Europe. However, I believe many of the methods used are inconsistent with these other studies and are not sufficiently justified or argued for in the text. All the methods used are slightly different from those established in the literature, such as tracking at 700 hPa instead of 850 hPa, analysing wind gusts at 100 metres instead of 10 metres, and using an extremely high exceedance threshold of the 99.9th percentile instead of the more commonly used 98th percentile. I would like to see some evidence for the choice of the methods. Furthermore, the introduction and framing of the paper (especially the first half) feels very incoherent and I believe requires significant re-structuring. In addition, the presentation of the figures feels clumsy and very difficult to interpret with numerous similarly styled lines and colours overlayed on similarly shaded fields, which needs rectifying. My full points can be found below. I therefore recommend that this study requires major corrections.

**Please see our comments above regarding use of 10 m v 100 m and sustained wind speeds v wind gusts. Additionally, the reviewer mentions use of RV at 850 hPa in prior research over northern Europe for cyclone tracking. This level is not appropriate for use in North America because of the high terrain elevation that means the 850 hPa is frequently below the land surface. It is deemed preferable by the authors to use a level that is more typically realized in the atmosphere. Finally, we regret you had difficulty reading the figures, and have modified some as appropriate, and enlarged those that we could, to aid clarity.**

Individual points

1.      L43 – I feel the reference here to Fig. 1a should illustrate the two different types of cyclone tracks you are describing. As the figure is for wind gusts it in no way illustrates the differences in the genesis location

**The Purpose of Figure 1a Is to explicitly communicate the locations of the Northeast states to readers less familiar with U.S. geography. The state names and abbreviations are used throughout the paper. We believe that changing the scale of this map to include cyclone genesis locations would make the state borders and labels difficult to interpret. The classification of storm tracks into; Alberta Clippers, Colorado Lows and tropical cyclones is consistent with past cyclone climatologies and is described at length in the introduction (lines 34 to 54)**

2.      L47-49. How is the influence of post-tropical cyclones in the USA consistent with Europe? Are less than 1% of cyclones also post-tropical in this region? Perhaps it is best to remove the 1% statement.

**We regret the ack of clarity and have re-worded to; 'Research on windstorm risk in Europe found that although less than 1% of cyclones that impact Northern Europe are post tropical cyclones, they tend to be associated with higher 10-m wind speeds (Sainsbury et al., 2020). Tropical cyclones are a major driver of extreme wind speeds along the U.S. eastern seaboard (Barthelmie et al. 2021) and events such as Hurricane Sandy have been associated with large geophysical hazards in the Northeast (Halverson and Rabenhorst, 2013;Lackmann, 2015).'**

3.      L61-63. Can a trend line or some evidence of the trend from the data in Fig. 1b also be included/quoted? Due to the large inter-annual variability of the data it is hard to tell from the figure you have presented that there is a positive trend. Furthermore, you quote the 98th percentile in the text (perhaps this refers to the results of Bronnimann et al., 2012), yet the figure in question (I assume Fig. 1b?) uses number of grid points exceeding the 99.9th percentile of U. Please clarify these differences,

or make the figure consistent with the text.

**We believe the reviewer was confused by the reference to Figure 1 and Table 1 that emphasized/described the GEOGRAPHICAL region. We have removed those to avoid the possibility of confusion.**

4.        Section 1.3. From your introduction there is limited evidence of where this study fits within the established research and scientific literature. I think it could be framed better to give a clear narrative as to where the gaps in the literature are and how this study addresses those gaps.

**We are very explicit in our objectives and links to past research have been clarified and extended in the revised version of the manuscript.**

5.        L123-124.  I feel this statement is incorrect for the last 10-15 years with the introduction of more advanced and homogenous reanalysis products. Please make it clearer how ERA5 is beneficial compared to previous generation products such as ERA-Interim, etc.

**We believe the real advantages of ERA5 are those we state in section 2.1. Assimilation of a massive suite of data; 'The ERA5 reanalysis is derived using an unprecedented suite of assimilated in situ and remote sensing observations (Hersbach et al., 2020). '. Its relatively high demonstrable skill for wind speeds; 'It exhibits relatively high fidelity for wind speeds (Kalverla et al., 2020;Olauson, 2018;Kalverla et al., 2019;Pryor et al., 2020;Jourdier, 2020;Ramon et al., 2019)'. AND the relatively high resolution; 'a spatial resolution of 0.25°×0.25°'. This latter point – as the reviewer will know, compares very favorably with past global reanalysis products. Eg.;  MERRA-2; 0.625°×0.5°, ERA-Interim (approximately 80 km).**

6.        I am a little confused as to why the 100 metre level is being used throughout this study as you are specifically interested in damaging windstorms. You have justified this in the text, however I feel it may be better to be consistent with previous studies, which generally use the 10 metre level. Does using 100 metres lead to drastically different results to using 10 metres? Furthermore, as you are using near-surface ground observations as a validation, surely it would make sense to also use the 10-metre product from ERA5 as there could be significant differences in the 100m and 10m distributions.

**Please see above.**

7.        As above,  doing the tracking at 700 hPa  instead of 850 hPa  is also confusing to  me. Using the Hodges method tracking is mostly done at 850 hPa. Are there specific issues with representing the flow in the orographic regions? Are all cyclones identified at 700 hPa the same as those that would be identified at 850 hPa?

**Please see above. In the text we note this justification briefly where we state; 'RV values at 700 hPa are used rather than 850 hPa as in the XWS European analysis due to the presence of high elevation areas in U.S. cyclone source regions.'**

8.        I feel the methodology section can be condensed significantly. You introduce the tracking on line 148 and then describe it in detail, several pages later. It feels very incoherent and should be re-structured for  conciseness.

**We briefly introduce use of the ERA5 data for the cyclone tracking in section 2.1 and then present the methodology in section 2.5. We believe that is appropriate, but have edited the Data and methods section for brevity.**

9.        L210 – you use the 99.9th percentile, what is the justification for this? As you mentioned in the text your method is similar to numerous other storm severity assessments, however most use the 98th percentile. Due to the relative short extent of data from ERA5, are there not chances that the 99.9th percentile is exceptionally skewed by very large  events?

**This is an important point. We have amended the introduction to read; 'This research is a part of the HyperFACETS project which uses a storyline-based analysis framework. Storylines are "physically self-consistent unfolding of past events, or of plausible future events or pathways" (Shepherd et al., 2018).**

**They provide a method of framing a research inquiry in terms of three elements: A geographic region, a historically important or notable event, and a set of process drivers for that event.' To clarify we are indeed seeking to identify and characterize truly exceptional events.**

10.  L223 – is it possible that by using such a high threshold some spatially large events (p98<U<p99.9) are missed and events with a very small area of U>99.9 are counted instead? It may be that in some of these cases could the large, yet slightly lower intensity events have larger impacts than small scale high-wind events.

**This is an important point. Use of any threshold inherently dictates the type of event that will be identified. As shown in Table 1 none of our events are confined to very small areas. Indeed, all of them cover at least a third of land grid cells. Nevertheless, we have added a discussion regarding the link between our events and their ranking using a different threshold (p98).**

11.  L225 – a 14 day restriction seems rather large. Have you tested this criteria to see if any high impact storms are excluded as a result of this threshold? Several studies have shown that more intense cyclones are more likely to cluster (e.g. Mailier et al., 2006), therefore this could be removing events from your set.

**The 14-day exclusion window is used to ensure that all storms on the top ten list are independent of one another, and this window is reduced to two days when the serial correlation of storms is considered. There is one storm which would have been included in the top ten, were it not for this 14-day window. This is noted in the new text:**

**"A larger sample of 50 windstorms was also drawn from the 40-year time series to examine the serial dependence. This analysis reduces the 14-day exclusion window used in the identification of the top 10 windstorms to a 2-day window. One windstorm (on January 19th, 1996) is excluded by use of a 14-day separation window from the list of the top ten storms but is included if a 2-day exclusion period is used. It would have been ranked number ten."**

12.  L235 – further references are needed here such as Vitolo et al. (2009), Mailier et al. (2006), Pinto et al. (2014).

**We understand there have been other articles published on serial clustering of windstorms in Europe. Given our work is focused on North America and we already have 109 references, we think it is sufficient to refer to one of those studies.**

13.  L265 – do you have specific requirements for each of these cyclone classes (i.e. genesis location). If so please state this in the text.

**We regret this lack of clarity. We have added the following text; 'A cyclone is identified as an Alberta Clipper (AC) if the cyclone track originates over the North American continent north of 40°N, as a Colorado low (CL) if the track originates over the North American continent south of 40°N, and as a decaying tropical cyclone (TC) if the track originates south of 30°N over a water grid cell. The term nor'easters (NE) is applied if the cyclone retrogrades towards the coastline after moving offshore and/or is associated with strong northeasterly flow over the Northeastern states.'**

14.  Fig 2. I find the layout and organisation of this figure very messy. The legends should be moved outside of the panel boundaries and all text on the panels be made clearer as it is very hard to decipher any of the information.

**We regret you had difficulty reading this figure. We have made it much clearer by moving the legends, removing damage estimates (since they are in the table) and increasing the size of the figure.**

15.  L317 – reference should be outside of brackets.

**Typographic error corrected.**

16.  L348-349 – as discussed above. If your systems traverse your region of interest in ~72 hours, why the 14 day separation? Please clarify this.

**We did not know before we did the analysis how long any individual storm would take to traverse the region. We wanted to set a sufficiently large window to ensure independence.**

17.        Fig 4 and throughout – is it worth referring to each storm by its ranking in the text    and figures.  I find it hard to keep track of which year is which through the text, this may   be a simple  way to avoid this.

**We think that using the consistent color scale is effective in this regard but have added some information about ranking to the text also.**

18.        L396-406 – would the authors be able to illustrate these results in some way instead of just giving a description.  Furthermore,  commonly dispersion is calculated  as counts per month, or counts per winter season, and not counts per calendar year. Vitolo et al. (2009) Fig. 15 demonstrates how dispersion (and surrounding uncertainty) commonly increases with aggregation period and as these storms you are interested  in are only features of approx. 6 months of the year is it likely that this dispersion value is representative?

**Possibly. Given the D value is small we prefer not to expand the discussion greatly.**

19.        Figs 6 and 7 – legibility of these figures is very difficult. Often green lines are plotted above an area of green (especially figure 7a) and also with text on the figures it makes it very difficult to distinguish and correctly identify features. I would recommend a redesign of these figures to aid legibility.

**Thank you, the background colors in these figures (now combined into Figure 6) have been changed to reduce confusion, and the track outline color has been changed from white to black. The overall size of the figure has also been increased.**

20.        Fig 7b and table 3 – are the mslp units in hPa the anomaly from the background field? This needs to be made clearer as the magnitude of the values are confusing.

**We have clarified by adding; 'These are anomalies identified in the filtered fields. RV cyclone intensities are shown in units of $10^{-5}$ s$^{-1}$, and MSLP intensity estimates are given in hPa scaled by -1. These anomalies are relative to removal of the large-scale background for n ≤ 5, where n is the total wavenumber in the spherical harmonic representation of the field.'**

21.        L412 – again, do you have requirements for these cyclone classes. It would be useful to define these earlier in the text instead of just approximating by eye if they are one class or the other. This is also applicable for L416 where you state a cyclone transitions to the NE  class.

**Please see response above.**

22.        L464-465 – please display this information somewhere in either a figure or table, with it being stated as is it feels unjustified. Also you state how all storms have RP >50 years in some location, is this evident in figure 8 as the colorbar does not extend beyond 50? Perhaps to make this clearer the authors should extend the colorbar beyond 50 years and then highlight the regions which exceed 50 years.

**The extent of the color scale in all panels of Figure 8 (Now Figure 7) has been increased to 60 years.**

23.        Figure 8 and table 3 – what is the uncertainty on the return period calculations? As with ERA5 there is only 40 years of data the uncertainty must be very large for the 100 year event. Please clarify and quantify this in the text and figures.

**Quite. We have added this comment to methods; 'Uncertainty intervals on the return period wind speeds are assigned using the 95% confidence intervals on the $\alpha$ and $\beta$ parameters as derived using maximum likelihood estimation.' And have added results to Table 3.**

24.        All figures – the visualisation and colour clashes at times makes for figures that are very

hard to interpret. Fig. 8 the red lines of states over the map is almost impossible to clearly distinguish. Fig 6 and 7 the track lines  an intensity circles are  hard to see   as they overlap and also clash with the background colours. Figure 2 has legends overlapping figure space and also text on figures that cannot be read. I feel a redesign of these figures is required to accurately present the authors results.

**All figures have been made larger to aid visibility. The red lines in Figure 8 (now Figure 7) have been changed to white to reduce visual clutter. Background colors in Figures 6 and 7 (now combined into Figure 6), have been changed to make the tracks, track labels, and intensity markers much more legible.**

25.          The storm severity metrics (following Klawa and Ulbrich, 2003) are introduced but in no way used in the analysis and I feel could be a strong contribution to the results. Performing an SSI-like calculation to compare ERA5 ranking to actual ranking would    be a useful addition to the analysis. It would be good to compare the U_999 spatial extent and maximum wind speed to this storm loss metric.

**At the reviewers request we have added a Loss Index based on their work in Europe (introduced in section 2.7) and results described in section 3.4. Our results suggest that for these ten windstorms the Loss Index is not very highly predictive of NOAA storm damages.**

References

Mailier, P. J., Stephenson, D. B., Ferro, C. A., & Hodges, K. I. (2006). Serial clustering of extratropical cyclones. Monthly weather review, 134(8), 2224-2240.

Pinto, J. G., Gómara, I., Masato, G., Dacre, H. F., Woollings, T., & Caballero, R. (2014). Large‐scale dynamics associated with clustering of extratropical cyclones affecting Western Europe. Journal of Geophysical Research: Atmospheres, 119(24), 13-704.

Vitolo, R., Stephenson, D. B., Cook, I. M., & Mitchell-Wallace, K. (2009). Serial cluster- ing of intense European storms. Meteorologische Zeitschrift, 18(4), 411-424.

Anonymous Referee #2

Review of "Windstorms in the Northeastern United States" by Frederick W. Letson, Rebecca J. Barthelmie, Kevin I. Hodges and Sara C. Pryor

The paper presents an analysis of the 10 largest windstorms over the northeastern United States during the past four decades. It combines metrics of extreme winds and precipitation based on both reanalyses and observations with tracks of the associated extratropical cyclones based on both pressure and vorticity. The time evolution of reanalysis and observational data qualitatively agrees during the storm events. The storms show typical tracks for the region but intensity about one magnitude higher than average. Several storms are associated with extreme damages over 1B$ and their winds show long return periods above one century locally.

The regional focus on the northeastern United States complements earlier windstorm studies that focus on northwestern Europe mostly. In that sense, the paper is an important contribution to the windstorm community. However, it suffers from several short- comings that limit its actual impact. The paper tends to cover too many topics without a clear focus and to combine too many approaches applied in an unusual way. It would benefit from a well-defined scope, standard methods taken from the literature and a better structure altogether. General and specific comments are listed below to help improve the paper quality.

GENERAL COMMENTS

I.        The scope of the paper is vague: it appears to be the 10 most intense (or largest?) windstorms over the 40 year period 1979–2018 but this is not clearly stated or motivated. Ten is not enough for a catalog and not significant to represent severe storms but likely too much for detailed case studies. The authors somehow need to choose between increasing the sample and focusing on a few particular storms.

**Our work is being conducted within a project that centers on development and use of storylines. As our modified draft now reads; 'This research is a part of the HyperFACETS project which uses a storyline-based analysis framework. Storylines are "physically self-consistent unfolding of past events, or of plausible future events or pathways" (Shepherd et al., 2018). They provide a method of framing a research inquiry in terms of three elements: A geographic region, a historically important or notable event, and a set of process drivers for that event.'**
**We hope this rewording helps to clarify. For the purposes of clarifying our purpose to the reviewer we note use of storylines facilitates interactions with stakeholders (i.e. they likely remember these extraordinary events) and also enables future work – e.g. using pseudo-global warming simulations to examine how such events might evolve in the future.**

II.        The methods are comprehensive (4 pages of description) but uncommon and not sufficiently explained or motivated: 100 m rather than 10 m winds, 99.9th rather than 98th percentile, intensity defined as instantaneous spatial extent of extreme wind rather than the footprint of, e.g., cubed winds above some threshold. This may result in the obtained ranking of storms not matching their damage.

**Please see response above. We totally understand the reviewers (and editor) have conducted previous research over northern Europe that used a different methodology but feel confident ours is 'fit for purpose' for example it does identify windstorms that are present in the memories of our stakeholders.**

III.        Analyzing compound events is certainly meaningful but throughout the paper it is unclear whether the damages associated with the 10 selected storms are due to wind, precipitation, or both. This is somehow linked to the comment above, which applies to precipitation metrics as well.

**We completely concur. In our new section 3.4 (added at the request of reviewer 1) we discuss this in more detail.**

SPECIFIC COMMENTS

l. 1 The title should be more specific, e.g., referring to the most severe windstorms in the region

**We have modified the title to be; Intense windstorms in the Northeastern United States**

l. 14–15 "Alberta Clippers", "Colorado Lows" and "Nor'easters" sound too specific for the abstract.

**We think linking to the cyclone responsible for the windstorms is useful so, with apologies, invoke authors privilege.**

l. 19 Why "those windstorms that occurred after the year 2000" only? What about the others?

**The National Weather Service ASOS network of meteorological stations was subject to a series of notable upgrades that were concluded in 1999 (e.g. installation of sonic anemometers). Thus, verification of ERA5 winds and precipitation are most robust after the year 2000.**

l. 28 In the midlatitudes?

**Quite possibly, but as we exclusively cite studies from north America and Europe, we feel it is better to be specific.**

l. 36–37 The second part of the sentence appears unnecessary.

**We believe that the reviewer is referring to 'It lies under a convergence zone of two prominent Northern Hemisphere cyclone tracks associated with cyclones that form or redevelop as a result of lee-cyclogenesis east of the Rocky Mountains' We express it in this way because some weak cyclones do traverse the Rocky Mountains and re-intensify.**

l. 43 What is to be seen on Fig. 1a?

**We are notifying readers who may not know where Lake Superior is that they can see it in Figure 1a.**

l. 48–49 I do not fully understand the sentence and the cited paper describes Europe only

**We regret you did not understand this sentence. We have re-worded to read; 'Research on windstorm risk in Europe found that although less than 1% of cyclones that impact Northern Europe are post tropical cyclones, they tend to be associated with higher 10-m wind speeds (Sainsbury et al., 2020). Tropical cyclones are a major driver of extreme wind speeds along the U.S. eastern seaboard (Barthelmie et al. 2021) and events such as Hurricane Sandy have been associated with large geophysical hazards in the Northeast (Halverson and Rabenhorst, 2013;Lackmann, 2015).'**

l. 75–99 (Section 1.2) Many studies and numbers are cited but they lack focus on the topic of the paper: windstorms over the northeastern US

**We regret you disagree with our choice of literature. Reviewer 1 suggested we add more literature about windstorms in Europe so it is indeed a difficult balance to strike. Compared to northern Europe there is little prior research on windstorms in the Northeast and we believe we have cited the majority of it.**

l. 86 Repetition of l. 33–34

**Quite correct, we apologize and have modified the text to avoid this duplication.**

l. 101–102 This requires more details here and should somehow appear in the title and abstract

**We believe the reviewer is asking us to add further details regarding 'This research is inspired by and is conceptually analogous to development of the XWS (eXtreme WindStorms) catalogue of storm tracks and wind-gust footprints for 50 of the most extreme European winter windstorms (Roberts et al., 2014).' We think that would not be appropriate to the abstract of our work.**

l. 105 Most intense or severe?

**Yes, this is an interesting point. We decided on intense because we cannot say they are the most severe given we use a relative threshold to absolute magnitude to define them.**

l. 117–120 This paragraph is unclear and seems misplaced

**We regret you felt so. We wanted to explain why we select extraordinary events.**

l. 120 A description of each Section is expected here

**It is not required by the journal style and checked five recently published articles in this journal (Nat. Hazards Earth Syst. Sci., 21, 587–605, Nat. Hazards Earth Syst. Sci., 21, 577–585, Nat. Hazards Earth Syst. Sci., 21, 607–627, Nat. Hazards Earth Syst. Sci., 21, 481–495, Nat. Hazards Earth Syst. Sci., 20, 3521–3549). Only one (Nat. Hazards Earth Syst. Sci., 21, 607–627) had such a statement so it appears to be at the discretion of the authors. We prefer not to include it as a matter of style since it uses space without conveying essential materials so for the time being, we do not include it.**
l. 121ff (Section 2) The study period is unclear and inconsistent between subsections

**We regret we are a little unclear on what the reviewer is referring to 1979-2018 is the time period we used exclusively to define the windstorms. Due to the presence of enhanced measurement system prior to 2000 the detailed evaluation is performed for a subset of the windstorms that occurred after 1999.**

l. 123 Over long periods? It is not an issue for case studies

**We have reworded to 'Attempts to identify and characterize windstorms from a geophysical perspective and contextualize them in a climatological context have historically been hampered by limited data availability and/or quality from geospatially inhomogeneous observing networks.'**

l. 128–129 Hourly or every 20 minutes?

**We have reworded to; 'Thus, herein we employ once-hourly wind speeds from the ERA5 reanalysis. The wind speeds are for a height of 100-m a.g.l. at the model time step of 20 minutes and a spatial resolution of 0.25°×0.25°.'**

l. 137–144 The motivation for using winds at 100 m agl is not fully convincing and slightly repetitive; this height is used for wind power mainly and may not reflect the surface impact of windstorms

**We have rewritten to; 'Here we use wind speeds at 100-m because flow at this height is less likely to be impacted by sub-grid scale heterogeneity in surface roughness length and uncertainties induced by unresolved sub-grid scale variability. Near-surface wind speeds are strongly coupled to wind speeds at 100-m (i.e. within the PBL) but wind speeds at 100-m are less strongly impacted by inaccuracies and/or uncertainty in surface roughness length ($z_0$) (Minola et al., 2020;Nelli et al., 2020). Applying an uncertainty of a factor of two to $z_0$ can lead to mean differences of up to 0.75 ms$^{-1}$ for near-surface (40 to 150 m a.g.l.) wind speeds (Dörenkämper et al., 2020). Further, the scale of events we seek to characterize are regional rather than local scale, and are necessarily driven by winds aloft.'.**

l. 145 Why 3-hourly rather than hourly as above?

**Because its highly redundant. Most prior research has used 6-hourly (e.g XWS).**

l. 155 Which ten storms?

**Reworded to; 'Precipitation intensity and hydrometeor class from ERA5 are used to identify to what degree each of the ten windstorms identified here are compound events.'**

l. 168–171 I agree but this does not support the use of 100-m winds from ERA5

**That is right. We believe from a theoretical perspective using 100-m a.g.l. wind speeds IS preferable (see discussion above). However, we have to be pragmatic about data availability for evaluation.**

l. 175 Twice "product(s)"

**Apologies. Corrected**

l. 179–183 The two sentences sound repetitive

**Quite. Corrected to; 'In the current work, precipitation rates over the land areas of Northeastern states from RADAR and ASOS and ERA5 that are within 200 km of the 7 RADAR are averaged in time to match the hourly resolution of ERA5 precipitation and interpolated in space to the 0.25°×0.25° ERA5 grid. (Fig. 1c).'**

l. 193–195 Verb missing?

**No but the 'to' was extraneous. Apologies.**

l. 194 What is the caveat here? The impact is indeed expected to depend on population density

**Yes, our meaning was lost here. Reworded to; Damage and mortality estimates from this dataset provide an estimate of the impact of each windstorm, with the caveat that population density and hence the potential for loss of life and damage vary markedly between U.S. counties that also vary greatly in area (Fig. 1d).**

l. 210–212 This approach reflects the severity rather than intensity (storm severity index)

**See response above.**

l. 212–215 The motivation for using the 99.9th percentile is unclear; although the approach is disputed by other authors, the choice of the 98th percentile arises from comparison with impact data in Klawa et Ulbrich (2003)

**We hope we have clarified this matter in our other responses.**

l. 215–217 Why?

**We believe the reviewer is referring to; 'While lower percentile thresholds have been used in previous work (Walz et al., 2017;Klawa and Ulbrich, 2003), use of the 99.9$^{th}$ percentile wind speed value is appropriate for identifying the truly extraordinary conditions we seek to characterize and is robust when applied to very long datasets with very large sample sizes.'**

**We really do seek to identify truly exceptional conditions and as we now show use of 98$^{th}$ percentile is not very discriminating.**

l. 225 14 days appear as a very long period to separate events

**Possibly but we wanted to ensure that Nor'easters' were not double counted.**

l. 227 if values are instantaneous then +/-48h makes 96h

**Right but as we can see none last the full time window. The data used represent 97 hours.**

l. 235–236 The focus on the 10 most intense (severe?) storms should be stated clearly and early in the paper

**It is mentioned in the objectives explicitly.**

l. 244 Reference without brackets

**Corrected.**

l. 259–266 Do the centroids of extreme ERA5 winds actually coincide with extratropical cyclones, in general and for the 10 most intense cases in particular?

**No – this is shown in Figure 2.**

l. 267–268 If I understand correctly the intensity is solely defined as the maximum instantaneous areal extent of extreme winds?

**Yes.**

l. 299–305 (Table 2) The windstorm category would be useful here (AC, CL, TC, NE)

**We think it is more appropriate in Table 3.**

l. 306–311 What are the actual most damaging extratropical cyclones for the considered period, and are they captured by the selection?

**We think the reviewer is referring to Table 2. Certainly, we identify events that stand out in the damage record but in this region flooding linked to storm surge, and heavy precipitation also yield some high NOAA storm damage reports. Perhaps this is not what the reviewer is asking. We can say these ETC are exceptional relative to others with similar tracks see Figure 6 and associated text.**

l. 313–320 An additional but crucial explanation may be that the used metric is simply not appropriate… In addition to different wind height and percentile, standard metrics are based on a footprint and of some function of winds (typically cubed) rather than on an instantaneous spatial extent.

**Possibly – we have further elaborated on these matters in the new section 3.4 added at the request of the other reviewer.**

l. 334–336 The strongest winds often occur to the south of the pressure center but not necessarily related with the cold front (see, e.g., the Hewson and Neu 2015 paper cited above)

**Quite. We were too brief here we added an additional reference that talks more about some of the other mechanisms.**

l. 340–345 (Figure 2) What is the reason for the large discrepancy between RV and MSLP tracks in some cases?

**This response is combined with the response to the nest comment.**

l. 350–351 It would be interesting to know why it is the case (1981 and 2003 storms)

**We have added text to section 3.2 that addresses this matter; For most cyclones independent tracking of the center using MSLP and RV yields results that are highly consistent (Fig. 2). Nevertheless, some discrepancies exist in part due to the spectral field smoothing. Another possibility is if there is a strong background flow due to a strong pressure gradient, the vorticity can be offset relative to the pressure minimum (Sinclair, 1994).**

l. 362 Hurricane Sandy was already introduced

**Deleted.**

l. 363–364 How many is "several" or "multiple" grid cells? 20 mm in 24 h is far from extreme. . .

**Thank you. This text has been updated to read:**

**"Hurricane Sandy (windstorm during 2012) is associated with total 24-hour precipitation accumulation exceeding 100 mm in 0.5% of ERA5 grid cells, and exceeding 20 mm in 46% of ERA5 grid cells."**

l. 370 (Figure 4) The histograms do not emphasize extreme (i.e., potentially damaging) precipitation. And what is precipitation type 2?

**Thunderstorm. We did not refer to it because it was not present for any of our windstorms, but have added it to the caption at your request.**

l. 387–390 The underestimation of station observations may be explained by the spatial variability of precipitation but there is a factor ∼2 between ERA5 and radar estimates

**Quite. It is very interesting. We do not have an explanation but note this recent publication; Cucchi, M., Weedon, G. P., Amici, A., Bellouin, N., Lange, S., Müller Schmied, H., ... & Buontempo, C. (2020). WFDE5: bias-adjusted ERA5 reanalysis data for impact studies. *Earth System Science Data*, *12*(3), 2097-2120.**

l. 396–406 There is a confusion between clustering on yearly and daily time scales; I would expect the analysis focuses on the latter to distinguish between consecutive storms

**We regret the confusion and have reworded.**

l. 407ff (Section 3.2) This Section generally lacks structure

**We have reorganized this section.**

l. 408–412 How are these cyclones selected? Based on thresholds in MSLP or vorticity? The total number of extratropical cyclones likely exceeds 10 tracks per year by far

**Right. we tested different limits and found this to illustrate the tracks well. Note this figure has been redrafted at the request of Reviewer 1.**

l. 410 What are transitory cyclones?

**We are using it in the context of** transient weather systems, i.e. moving. Not, for example, the thermal low that dominates the desert Southwest.

l. 416–419 This was already stated above.

**We believe the reviewer may be referring to ; 'Consistent with past research employing other reanalysis data sets (Ulbrich et al., 2009), results from application of the cyclone detection and tracking algorithm to ERA5 output also indicate the U.S. Northeast exhibits a high frequency of transitory cyclones (Fig. 6). Also in accord with expectations, the tracks followed by the windstorms are generally characteristic of those dominant cyclone tracks, and derive from a mixture of intense nor'easters (NE), Alberta Clippers (AC), deep Colorado lows (CL), and decaying tropical cyclones (TC) (Table 3, Fig. 6).' We think it is worth linking our work to past research and expectations since we have not yet describe the cyclone climatology.**

l. 425–429 This belongs to the Introduction

**We believe the reviewer MAY be referring to; 'The Great Lakes are known to have a profound effect on passing cyclones during ice-free and generally unstable conditions that prevail during September to**

**November (Angel and Isard, 1997). Particularly during the early part of the cold-season, cyclones that cross the Great Lakes are frequently subject to acceleration and intensification via enhanced vertical heat flux and low-level moisture convergence due to the lake-land roughness contrast (Xiao et al. 2018). Cyclones such as Alberta Clippers that transit the Great Lakes during periods with substantial ice cover are subject to less alteration (Angel and Isard, 1997).' We concur and have moved this text.**

l. 436 The impact of the 2018 (not 2017) windstorm may be due to precipitation rather than wind mostly

**Thank you for catching that typo. Yes, it is possible. Please see the new section 3.4.**

l. 436–447 The impact is already discussed above (e.g., l. 418–424) and in other sections. Please reorganize

**Done.**

l. 450–460 (Figures 6–7) Same question as l. 340–345 (Figure 2)

**See response above.**

l. 463ff (Section 3.3) I am lost in the numerous details and wonder what to learn from this additional section

**We believe the reviewer is referring to the return periods analysis. They earlier asked do our analysis approach identify real extremes. We believe this section is crucial to answering yes!**

l. 496–502 This belongs to the Introduction

**We regret we are unsure what text the reviewer is referring to.**

l. 503 It is the first mention of the "largest" windstorms

**Quite- we regret this typo and have corrected it.**

l. 518–520 I do not agree on the accord with damage estimates

**We have reworded to; The statistically significant correlation between the ERA5 windstorm intensity estimates and independent damage estimates provides further confidence in the fidelity of the windstorm catalogue presented herein.**

l. 530–533 This is not convincing

**We wonder if the reviewer is referring to; 'These cyclones, however, exhibit considerably higher intensities (from both RV and MSLP perturbations) that are an order of magnitude higher than mean values sampled on those same tracks (Fig. 7). With the possible exception of Hurricane Sandy, these windstorms are largely differentiable from the cyclone climatology in terms of their intensification rather than the associated cyclone storm track.' We have reworded to; 'These cyclones, however, exhibit intensities (from both RV and MSLP perturbations) that are an order of magnitude higher than mean values sampled on those same tracks (Fig. 6). With the possible exception of Hurricane Sandy, these windstorms follow tracks that are not infrequent in the cyclone climatology.'**

l. 560ff (References) Please . . . increase . . . spacing . . . between . . . lines

**Done. Thank you.**

---

## Referee Report (RR1)

The authors have addressed the comments previously and I am satisfied with their justification of the methods used and also the additional included analysis regarding the p98 and p99.9 disparities. I still feel that the manuscript is in some places confusing and is not particularly easy to follow. Sections 3.2-3.4 I had particular difficulty following and a few specific points are detailed below. I would recommend the authors try to increase the clarity of this final part of the results as I believe it is very important in quantifying the differences in the impacts and loss potential for the different classes of cyclone. I am particularly happy with how the authors have re-phrased the introduction and methods of the paper. Below I have included some more detailed points that I would like to see addressed.

1. In the abstract it may be worth adding a statement as to the purpose of the study (similar to what is in lines 102-104)
2. Lines 21-22 and 440-443. You state that the value for D is less than in Europe, however the method used for the cyclone counts will likely provide a value of D that is different from the previous studies. Can you therefore state that the D value of 0.18 is less than what you would see over Europe using your method?
3. Lines 35-37: is it also possible to include a similar figure that is just a result of windstorms or extratropical cyclones? Or is annual data like this not obtainable?
4. Line 45: You state 'Previous research has found that these cyclones…' I assume this is referring to the Alberta Clippers, but could the authors please make this clearer in any potential cases that a reader thinks a Colorado Low and be north of Lake Superior.
5. Line 115. This last part of point 3 is a repeat of the first line of point 3.
6. Line 222-223: In the analysis do you apply any threshold for the wind speed exceedance being a set distance from the cyclone (You discuss how this is part of the XWS analysis on line 204). It is probably unlikely that due to the extreme threshold employed that there will be any exceedances that are not a result of the cyclone, but it should be mentioned whether or not any distance criteria is used.
7. As the comment above, are any thresholds used to associate precipitation to the cyclones, there may be considerable separation between the windstorm, cyclone, and maximum precipitation and is something that should be addressed.
8. Line 318-320: Would it be possible for the authors to put the figure from the first round of reviews in the appendix/supplement and reference here?
9. In Fig. 4 can the bins be changed for the max. precip charts? As so much of the data is skewed toward low precip rates some more detail max be gained from limiting the y-axis to 20 or 30 mm.
10. Line 440-445: Some of the studies you have referenced (especially Mailier et. al., 2006) show that D is often negative in parts of the northeastern USA, indicating regularity. Would the authors be able to hypothesise/discuss at all why these 50 storms indicate a clustered behaviour.
11. Figure 6. Is the heatmap of cyclone density over the NE USA just for cyclones that pass through your area of interest or is this a total climatology. This needs to be stated in either the caption or the text. I also find it very hard to identify the different sized circles in (c) and (d), as a suggestion it may be clearer to just include the circles when above the XXth percentile? Another suggestion for this figure as I

still ifnd the different coloured lines very hard to distinguish, would it be possible to have the different cyclone types as varying shades of the same colour?

12. Line 449: Please rephrase the start of the sentence 'Also in accord with expectations…' as it is unclear if you are referring to the climatology of tracks or your 10 intense windstorms.

13. Line 456: would you be able to quote some values here? E.g. most cyclones exceed 10x10**-5 whereas the climatology is approx. 1x10&**-5?

14. Line 461-462: I assume you mean that the NE USA is dominated by CL-type cyclones?

15. Section 3.2. One thing that may be discussed is what stage in the lifecycle are the cyclones at when they cause the most damage/are over the built-up regions. This is information that is included in Fig. 6 but not really discussed by the authors. This may be something that separates the high loss storms with those of lower losses.

16. Line 501: Is the median RP the median RP for all cells in the NE USA region? If so is this including a lot of grid points that are not part of the windstorm/cyclone area and substantially lowering the RP? This ties in a bit with points 6 and 7 above as to setting a cyclone area.

17. Section 3.4: I understand that the authors have added this in in response to the previous round of comments, however I feel it does not add any substantial discussion to the paper in its current form. Perhaps the figure included in the responses would be a better option to illustrate the differences between the loss model and the actual losses and the role of other hazards?

---

## Author Response (AR2)

**Response to referee comments for NHESS-2020-345**

We would like to thank both referees for their thoughtful contributions to this paper. Our responses to referee comments are shown below **in bold**.

**Referee 1**

Review for Letson et al.
The authors have addressed the comments previously and I am satisfied with their justification of the methods used and also the additional included analysis regarding the p98 and p99.9 disparities. I still feel that the manuscript is in some places confusing and is not particularly easy to follow. Sections 3.2-3.4 I had particular difficulty following and a few specific points are detailed below. I would recommend the authors try to increase the clarity of this final part of the results as I believe it is very important in quantifying the differences in the impacts and loss potential for the different classes of cyclone. I am particularly happy with how the authors have re-phrased the introduction and methods of the paper. Below I have included some more detailed points that I would like to see addressed.
**Thank you for your thoughtful comments and suggestions.**

1. In the abstract it may be worth adding a statement as to the purpose of the study (similar to what is in lines 102-104)
**Thank you. The following text has been added to the Abstract:**
**"The objective of this study is to identify and characterize intense windstorms during the last four decades in the U.S. Northeast and determine both the sources of cyclones responsible for these events and the manner in which those cyclones differ from the cyclone climatology."**
2. Lines 21-22 and 440-443. You state that the value for D is less than in Europe, however the method used for the cyclone counts will likely provide a value of D that is different from the previous studies. Can you therefore state that the D value of 0.18 is less than what you would see over Europe using your method?
**This is a good point. We have changed the wording to reflect this.**
**The last sentence of the abstract now reads:**
**"A larger pool of the top 50 largest windstorms exhibits evidence of only weak serial clustering which is in contrast to the relatively strong serial clustering of windstorms in Europe."**
**The final sentences of Section 3.1 now reads:**
**"While this D value (0.18) is symptomatic of serial clustering for windstorms that impact the Northern USA. Much higher serial clustering was reported for regions of European in earlier research using the 20th century ERA reanalysis and a 98th percentile wind speed threshold (Walz et al., 2018). The lower amount of serial clustering of windstorms in the Northeastern states at the annual timescale is indicative of a lower probability of multiple damaging windstorm events occurring within a single year."**
3. Lines 35-37: is it also possible to include a similar figure that is just a result of windstorms or extratropical cyclones? Or is annual data like this not obtainable?
**Unfortunately, the manner in which the NOAA storms report encodes the data precludes this.**
4. Line 45: You state 'Previous research has found that these cyclones…' I assume this is referring to the Alberta Clippers, but could the authors please make this clearer in any potential cases that a reader thinks a Colorado Low and be north of Lake Superior.
**Thank you. In response to this comment and a similar one from Referee 2, this has been edited to read:**

**Alberta Clippers generally move southeastward from the lee of the Canadian Rockies toward or just north of Lake Superior (Fig. 1a) before progressing eastward into southeastern Canada or the northeastern United States, with less than 10% of the cases in the climatology tracking south of the Great Lakes (Thomas and Martin, 2007).**

5. Line 115. This last part of point 3 is a repeat of the first line of point 3.

Thank you for noticing this. This has been edited to read:

**"3) Contextualize these windstorms in the long-term cyclone climatology. Specifically, we track each windstorm over time and space using two indices of intensity derived from mean-surface pressure and relative vorticity and compare their tracks and intensities to those of all cold-season cyclones affecting the Northeastern US from 1979 to 2018."**

6. Line 222-223: In the analysis do you apply any threshold for the wind speed exceedance being a set distance from the cyclone (You discuss how this is part of the XWS analysis on line 204). It is probably unlikely that due to the extreme threshold employed that there will be any exceedances that are not a result of the cyclone, but it should be mentioned whether or not any distance criteria is used.

**No distance criteria is applied but as shown by the close tracking of the geographic centroid of the high wind region and the cyclone centroid the high wind speeds are associated with these cyclones. The statement on Line 204 has been expanded to read:**

**"This approach is conceptually similar to storm severity indices derived from European work based on the maximum 925 hPa wind speed within a 3° radius of the vorticity maximum and the area over which wind speeds at that height exceed 25 ms-1 (Roberts et al., 2014;Della-Marta et al., 2009), while the current work considers over-threshold winds within a fixed domain 15 × 25° in extent."**

7. As the comment above, are any thresholds used to associate precipitation to the cyclones, there may be considerable separation between the windstorm, cyclone, and maximum precipitation and is something that should be addressed.

**Thank you for noticing this omission.**

**The precipitation data summarized in Figure 5 are for grid cells which exceed their local intense wind threshold ($U_{999}$) at least once during the 24 hours surrounding the storm peak. Additionally, the time series in Figure 6 illustrate the time delay between peak wind speeds and peak precipitation. The caption of Figure 5 has been edited to make this clearer.**

**It now reads:**

**"Figure 5. Histograms of precipitation totals and maximum precipitation rates and precipitation types for the 24 hours centered on each storm peak. All ERA5 land-based grid cells in the Northeastern states which exceed their local $U_{999}$ value at any point in the 24-hour period are included. The frequencies are the fraction of such grid cells in each class."**

8. Line 318-320: Would it be possible for the authors to put the figure from the first round of reviews in the appendix/supplement and reference here?

**We believe the referee is referring to our scatter plots of 100-m winds and 10-m gusts for each storm. At their request we have added this figure. It is now Figure 2 in the Paper. Please note the later figure numbers have changed as a result.**

9. In Fig. 4 can the bins be changed for the max. precip charts? As so much of the data is skewed toward low precip rates some more detail max be gained from limiting the y-axis to 20 or 30 mm.

**Thank you for the suggestion. The axis limits and bins size for the Max Precip. Panels have been decreased to better show the range of values which occur in the data.**

10. Line 440-445: Some of the studies you have referenced (especially Mailier et. al., 2006) show that D is often negative in parts of the northeastern USA, indicating regularity. Would the authors be able to hypothesise/discuss at all why these 50

storms indicate a clustered behaviour.

**The likelihood is that it is linked to the action of climate modes e.g. the PNA that has some low frequency memory or the AMO which again has low frequency memory but we have not undertaken an analysis of this (yet) so can not (yet) offer a robust explanation.**

11. Figure 6. Is the heatmap of cyclone density over the NE USA just for cyclones that pass through your area of interest or is this a total climatology. This needs to be stated in either the caption or the text. I also find it very hard to identify the different sized circles in (c) and (d), as a suggestion it may be clearer to just include the circles when above the XXth percentile? Another suggestion for this figure as I still ifnd the different coloured lines very hard to distinguish, would it be possible to have the different cyclone types as varying shades of the same colour?

**We have remade the figure to address this. Markers are now excluded for cyclone intensities below the 50th percentile (considering all points in the top 10 storm tracks) and the range of sizes in increased. We have elected to keep the basic track colors as they are, because this array of colors is important to the clarity of Figure 4. In order to make the tracks easier to see, we have thickened the black borders on each, and lightened the background in panel c.**

**The following text has been added to the caption to make it clear that the heatmaps only include cyclones which enter the Northeast:**

**"Background cyclone densities and intensities include only cold-season storm tracks that enter the Northeast rectangle."**

12. Line 449: Please rephrase the start of the sentence 'Also in accord with expectations…' as it is unclear if you are referring to the climatology of tracks or your 10 intense windstorms.

**Thank you This has been reworded for clarity. It now reads:**

**"Also in accord with expectations, the tracks followed by the top-10 windstorms are generally characteristic of those dominant cyclone tracks, and derive from a mixture of intense nor'easters (NE), Alberta Clippers (AC), Colorado lows (CL), and decaying tropical cyclones (TC) (Table 3, Fig. 7)."**

13. Line 456: would you be able to quote some values here? E.g. most cyclones exceed 10x10**-5 whereas the climatology is approx. 1x10&**-5?

**Good idea. This passage now reads:**

**"Cyclone intensities for the top 10 windstorms are an order of magnitude above the mean intensities for cold-weather cyclones at the same locations over the U.S. for both RV and MSLP. The median intensity of RV tracks for the 10 storms is $7 \times 10^{-5}$ s$^{-1}$ as compared to $6 \times 10^{-4}$ s$^{-1}$ for all cold-season tracks affecting the Northeast. The median intensity of MSLP tracks for the 10 storms is 25 hPa as compared to 1.2 hPa for all cold-season Northeast storms (Fig. 7, Table 3)."**

14. Line 461-462: I assume you mean that the NE USA is dominated by CL-type cyclones?

**Yes.**

15. Section 3.2. One thing that may be discussed is what stage in the lifecycle are the cyclones at when they cause the most damage/are over the built-up regions. This is information that is included in Fig. 6 but not really discussed by the authors. This may be something that separates the high loss storms with those of lower losses.

**reworded to:**

**"The geographic centroids of high wind speeds track through Virginia, Pennsylvania, and New York in all three events, but the advection velocity of the cyclones and the point in their lifecycle varies. Accordingly, inflation-adjusted damage amounts range from \$24 million for the 1981 windstorm to \$2181 million for the 1996 windstorm (Fig. 3 and 7)."**

16. Line 501: Is the median RP the median RP for all cells in the NE USA region? If so is this including a lot of grid points that are not part of the windstorm/cyclone area and

substantially lowering the RP? This ties in a bit with points 6 and 7 above as to setting a cyclone area.

**There are 3 values given in this table – as we note in the caption; 'Median RP is the 50th percentile return period for maximum wind speed in each Northeastern grid cell during each storm period, while p95 is the 95th percentile RP. Also shown is the median RP for grid cells that exhibited U>U999 at the storm peak. All RP values include a 95% confidence interval in parentheses.'**

17. Section 3.4: I understand that the authors have added this in in response to the previous round of comments, however I feel it does not add any substantial discussion to the paper in its current form. Perhaps the figure included in the responses would be a better option to illustrate the differences between the loss model and the actual losses and the role of other hazards?

**Yes. This was indeed added at the request of the reviewers and was not our research focus – which was the geophysical description of the windstorms. We think given two qualified and engaged reviewers requested it, it is an appropriate addition. The figure we chose is the one that we think best represents a clear tie back to our work and figure 1 that shows population density. At the request of the other reviewer, we had added a little more information regarding the inclusion/exclusion of Hurricane Sandy. The response to reviewers will also continue to be associated with this manuscript and thus will be available for readers.**

**Referee 2**

Second review of "Intense windstorms in the Northeastern United States" by Frederick Letson, Rebecca J. Barthelmie, Kevin I. Hodges and Sara C. Pryor

The paper has improved in clarity and the comprehensive methods are now better explained, although they sometimes appear inconsistent but are somehow justified by the results. Altogether most of my previous concerns have been addressed. However, besides a few minor issues, I am still not convinced by a systematic link between the extent of extreme winds and property damage. Comments are listed below to further improve the paper.

General comment

I appreciate the comparison between 100m wind and 10m gust as well as between U999 and U98 and between extent and severity. I believe this is sufficient to document the approach to select windstorms but I suggest the authors should not try to prove it works in general. Several events involve intense precipitation and #1 event Hurricane Sandy further involves a storm surge thus wind alone will not explain the exceptional damage. This is discussed in the paper but rather late in the results whereas it appears obvious and contradicts the methodology of considering both wind and precipitation to account for compound events.

**Thanks for your comments and thoughtful suggestions.**

Detailed comments

l. 32 "it" experiences: the region?

**This has been edited to read:**
**"The Northeastern US experiences a relatively high frequency of damaging storms, in particular during the cold season."**

l. 40, 42, 44 repetition of "lee of Rocky Mountains"
**Thank you. This has been made more concise:**
**"They lie under a convergence zone of two prominent Northern Hemisphere cyclone tracks associated with cyclones that form or redevelop as a result of lee-cyclogenesis east of the Rocky Mountains (Lareau and Horel, 2012). The first is associated with extra-tropical cyclones that have their genesis in the lee of Rocky Mountains within/close to the U.S. state of Colorado and typically track towards the northeast (Colorado Lows, CL) (Bierly and Harrington, 1995;Hobbs et al., 1996). The second is characterized by cyclones that have their genesis in the lee of Rocky Mountains in/close to the Canadian province of Alberta and track eastwards across the Great Lakes (Alberta Clippers, AC).**

l. 45 "these": which ones?
**This has been edited for clarity to read:**
**"Alberta Clippers generally move southeastward from the lee of the Canadian Rockies toward or just north of Lake Superior (Fig. 1a) before progressing eastward into southeastern Canada or the northeastern United States, with less than 10% of the cases in the climatology tracking south of the Great Lakes (Thomas and Martin, 2007)."**

l. 65–70 this is still very debated!
**We concur and have clearly noted a caveat and that a single reanalysis ensemble is the origin of this statement, so we think it is not being overstated:**
**"While long-term trends such as this from reanalysis products are subject to the effects of changing data assimilation (Bloomfield et al., 2018;Befort et al., 2016;Bengtsson et al., 2004), the 56 member twentieth century reanalysis exhibits a positive trend in the 98th percentile wind speed over parts of the U.S. including the Northeastern states that are the focus of the current research (Brönnimann et al., 2012). "**

l. 232 why use a much shorter period for precipitation than wind?
**The storm period (+/- 48 hours from the time of peak coverage), is designed to be large enough to show the entire temporal extent of the widespread high winds in each storm. For precipitation, we are primarily concerned with icing and heavy precipitation concurrent with the storm peak.**

l. 301 layout issue
**Thank you. This has been corrected.**

l. 306–307 confusing: per definition U999 is exceeded during 0,1% of hours
**This is a statement about the co-occurrence of $U_{999}$ exceedance in many grid cells.**

l. 338–344 and compound events involving storm surge and intense precipitation
We have these points to the list.

**"(iv) Compound events involving heavy precipitation, icing or storm surge (e.g. Hurricane Sandy (Wang et al., 2014)), along with intense winds may be associated with increased damage."**

l. 344–349 these statistics are not too impressive and likely biased by the presence of an outlier (Sandy) within the 10 events

**The reviewer is quite right, its a complex issue and the sample size is small. We report the association between the physical metrics and socioeconomic "damage" in two locations;**
**First at the point the reviewer refers to where we previously had;**
**Nevertheless, although many factors dictate economic losses from windstorms, the Pearson correlation coefficient (r) between the number of grid cells with $U > U_{999}$ at $t_p$ and inflation adjusted property damage exceeds 0.66, and r between the maximum wind speed and inflation adjusted property damage is 0.56. For a sample size of 10, using a t-test to evaluate significance (Wilks, 2011a), these correlation coefficients differ from 0 at confidence levels of 95% and 90%, respectively. We have now added; an additional sentence; 'Excluding Hurricane Sandy, increases r between the number of grid cells with $U > U_{999}$ at $t_p$ and inflation adjusted property damage to 0.86.' So we believe this now fully allows the reader to make their own assessment regarding the degree of correlation. Our assertion is that the correlation is statistically significant, despite other drivers being relevant. Second, at the reviewers request we added information regarding their loss damage index (section 3.4). We have added an additional statement about the effect of Hurricane Sandy, as an outlier, on the relationship between the size of each storm (# of cells with $U>U_{999}$) and the loss index estimated using their approach at the end of the results section (in section 3.4):**
**"A substantial fraction of variability in economic losses associated with these ten very high magnitude and large spatial extent windstorms is thus not well described solely by $U_{999}$. This is partly due to co-occurrence of other geophysical hazards (including flooding due the composite nature of some of these events, see Fig. 5). For example, the 2012 storm (Hurricane Sandy, ranked #1 in this analysis) is associated with greater property damage than would be predicted by either the LI or number of cells exceeding $U_{999}$, due to damage from storm surge and related flooding (Xian et al., 2015). Excluding Hurricane Sandy, the $R^2$ value computed using equation (5) for a linear fit with zero intercept between the LI and the number of grid cells with $U > U_{999}$ at $t_p$ decreases to 0.16."**
**It is worth reflecting on the INCREASE in correlation (r) between NOAA property damage (inflation adjusted) and # grid cells above $U_{999}$ at $t_p$ caused by excluding Hurricane Sandy but that excluding this event causes a decline in $R^2$ for a zero intercept linear fit between the NOAA Storm damage and # grid cells above $U_{999}$ at $t_p$. This is in large part because of the manner in which $R^2$ must be computed for a zero intercept (our equation 5) – it is NOT equivalent to the correlation coefficient squared. It is useful to consider what we are assuming when we force a zero intercept. It is that NO damage can occur if no grid cell exhibits $U>U_{999}$, thus one could argue that the correlation (without that assumption) include substantial complementary information. We remind the reviewer that we chose a $U_{999}$ threshold to ensure we are examining truly exceptional geophysical events not to imply no damage is possible at lower wind speeds.**

l. 400–403 see comment on l. 338–344
**This section is specifically about characterizing precipitation.**

l. 457–459 repetition of l. 392–297

**Yes. We believe that the fact that these storms are tropical cyclones is relevant in both sections**

l. 459–461 largely due to a storm surge
**Yes. We believe that this point has now been sufficiently made in other places in the text.**

l. 510 typo: Great Lakes "1"
**Thank you. This has been corrected.**

l. 544–546 see comment on l. 338–344
**Agreed. This has been addressed in our response to your comment above. l. 338-344**

Tables: it takes some effort to identify the storms mentioned in the text as YYYY as they are written in the tables as MM/DD/YY and not listed chronologically
**Thank you. This has been corrected in Tables 2 and 3. The dates are new listed in YYYY/MM/DD format. Windstorms are listed in rank order of spatial extent in both tables.**

[revised manuscript text omitted]